# Spinal lumbar dI2 interneurons contribute to stability of bipedal stepping

Baruch Haimson[1], Yoav Hadas[1], Nimrod Bernat[1], Artur Kania[2], Monica A Daley[3], Yuval Cinnamon[4], Aharon Lev-Tov[1]*, Avihu Klar[1]*

[1]Department of Medical Neurobiology, IMRIC, Hebrew University – Hadassah Medical School, Jerusalem, Israel; [2]Institut de recherches cliniques de Montréal (IRCM), Montréal, Canada; [3]Ecology and Evolutionary Biology, University of California, Irvine, Irvine, United States; [4]Institute of Animal Science Poultry and Aquaculture Sci. Dept. Agricultural Research Organization, The Volcani Center, Rishon LeZion, Israel

**Abstract** Peripheral and intraspinal feedback is required to shape and update the output of spinal networks that execute motor behavior. We report that lumbar dI2 spinal interneurons in chicks receive synaptic input from afferents and premotor neurons. These interneurons innervate contralateral premotor networks in the lumbar and brachial spinal cord, and their ascending projections innervate the cerebellum. These findings suggest that dI2 neurons function as interneurons in local lumbar circuits, are involved in lumbo-brachial coupling, and that part of them deliver peripheral and intraspinal feedback to the cerebellum. Silencing of dI2 neurons leads to destabilized stepping in posthatching day 8 hatchlings, with occasional collapses, variable step profiles, and a wide-base walking gait, suggesting that dI2 neurons may contribute to the stabilization of the bipedal gait.

*For correspondence:
aharonl@ekmd.huji.ac.il (AL-T);
avihu@mail.huji.ac.il (AK)

Competing interest: The authors declare that no competing interests exist.

## Introduction

The spinal cord integrates and relays the somatosensory inputs required for further execution of complex motor behaviors. Interneurons (INs) that differentiate at the ventral progenitor domain, V3-V0, are involved in the control of rhythmic motor activity, alternating between the left and right limbs, as well as between the flexor and extensor muscles (*Lai et al., 2016*; *Osseward and Pfaff, 2019*; *Alaynick et al., 2011*). Some of the dorsally born INs, dI1, dI3, and dI6, migrate ventrally and are also assembled within circuitries that control motor activity (*Yuengert et al., 2015*; *Bui et al., 2013*; *Andersson et al., 2012*), while other dorsal progenitor neurons, dI4 and dI5, give rise to INs that mediate somatosensation (*Lai et al., 2016*).

dI2 neurons originate in the dorsal spinal cord. The progenitor pdI2 cells, topographically positioned between the adjacent dorsally located dI1 and ventrally located dI3 neurons, express Ngn1, Ngn2, Olig3, and Pax3 transcription factors (TFs). Early postmitotic dI2 neurons undergo ventral migration and are defined by the combinatorial expression of Foxd3$^+$/Lhx1$^+$/Pou4f1$^+$ TFs (*Alaynick et al., 2011*; *Morikawa et al., 2009*; *Francius et al., 2013*). Importantly, none of these TFs is specific to dI2 neurons; rather, their combinatorial expression defines dI2. The lack of dI2-specific cell fate markers and the dynamic expression of the above TFs in other INs causes ambiguity regarding the molecular profile and outcome of late postmitotic dI2 neurons. Using intersectional genetics, we have shown previously that dI2 neurons are commissurally projecting neurons (*Avraham et al., 2009*). The lack of dI2-specific TFs impeded their genetic targeting. Hence, little is known about the wiring and physiological function of dI2 neurons.

The maintenance of stability and the coordination, precision, and timing of movements are regulated and modulated by the cerebellum. Anatomical and electrophysiological studies of cats and rodents revealed two major pathways ascending from neurons in the lumbar spinal cord to the

cerebellum: the dorsal spinocerebellar tract (DSCT) and ventral spinocerebellar tract (VSCT). DSCT neurons are considered to relay mainly proprioceptive information, while VSCT neurons are thought to relay internal spinal network information to the cerebellum along with proprioceptive data (*Jankowska and Hammar, 2013*; *Spanne and Jörntell, 2013*; *Stecina et al., 2013*; *Jiang et al., 2015*). While subpopulations of DSCT neurons are genetically accessible (*Hantman and Jessell, 2010*), the genetic inaccessibility of VSCT neurons hinders efforts to reveal their actual contribution to the regulatory functions of the cerebellum in locomotion and other motor behaviors.

In the present study, we investigated the possible functions of dI2 neurons in chick motor behavior. Employing intersectional genetics in the chick spinal cord, we targeted dI2 neurons and found evidence implicating them in the control of stability during locomotion. There are several advantages to performing these studies in chicks. The patterning of neurons within the spinal cord (*Jessell, 2000*) and the spinocerebellar tracts (*Furue et al., 2010*; *Furue et al., 2011*; *Uehara et al., 2012*) is conserved between mammals and birds. In addition, chicks use bipedal locomotion (evolved in humans and birds) that can be examined soon after hatching. To decode the circuitry and function of spinal INs, we developed a unique circuit-deciphering toolbox that enables neuron-specific targeting and tracing of circuits in the chick embryo (*Hadas et al., 2014*), and we utilized kinematic analysis of overground bipedal stepping by the hatched chicks following silencing of dI2 neurons.

Our studies revealed that lumbar dI2 neurons receive synaptic inputs from inhibitory and excitatory premotor neurons (pre-MNs) and relay output to the cerebellar granular layer, pre-MNs in the contralateral spinal cord and the contralateral dI2 cells. In a kinematic analysis of overground stepping by posthatching day (P) 8 hatchlings after inhibition of dI2 neuronal activity by expression of the tetanus toxin light chain gene, the genetically manipulated hatchlings showed an unstable gait, demonstrating that dI2 neurons play a role in shaping and stabilizing the bipedal gait.

## Results

To define potential VSCT neurons within spinal INs, we defined the following criteria: (1) soma location in accordance with precerebellar neurons at the lumbar level, which were previously revealed by retrograde labeling experiments of the chick cerebellar lobes (*Furue et al., 2010*; *Furue et al., 2011*; *Uehara et al., 2012*), (2) commissural neurons, (3) excitatory neurons, and (4) non-pre-MNs (*Lai et al., 2016*; *Osseward and Pfaff, 2019*; *Alaynick et al., 2011*). Based on these criteria, dI1c and dI2 neurons are likely candidates (*Bermingham et al., 2001*; *Yuengert et al., 2015*; *Figure 1—figure supplement 1*). This is further supported by *Sakai et al., 2012*, who demonstrated that in the embryonic day (E) 12 chick, the dI1 and dI2 axons project to the hindbrain and toward the cerebellum. In the current study, we focused on deciphering the circuitry and function of dI2 neurons and their possible association with VSCT.

### dI2 INs are mainly excitatory neurons with commissural axonal projections

To label dI2 neurons, axons, and terminals in the chick spinal cord, we used intersection between enhancers of two TFs expressed by dI2 – Ngn1 and Foxd3 – via the expression of two recombinases (Cre and FLPo) and double conditional reporters (*Figure 1—figure supplement 1A*). We have shown previously that the combination of these enhancers reliably labels dI2 neurons (*Avraham et al., 2009*; *Hadas et al., 2014*). The recombinases and the double conditional reporter plasmids were delivered via spatially restricted electroporation to the lumbar spinal cord at HH18. At E5, early postmitotic dI2 neurons migrate ventrally from the dorsolateral to the midlateral spinal cord (*Figure 1A*). As they migrate ventrally, at E6, dI2 neurons assume a midlateral position along the dorsoventral axis (*Figure 1B*). Subsequently, dI2 neurons migrate medially, and at E17, comparable to postnatal day 4 (P4) of mice, most of them (70.7 and 71.5 % at the sciatic and the crural levels, respectively) occupy lamina VII (*Figure 1C and F*). At all rostrocaudal levels and embryonic stages, dI2 axons cross the floor plate (*Figure 1A–D*). After crossing, dI2 axons extend rostrally for a few segments in the ventral funiculus (VF) and subsequently turn into the lateral funiculus (LF) (*Avraham et al., 2009*; *Figure 1C and D*). Collaterals originating from the crossed VF and LF tracts invade the contralateral spinal cord (*Figure 1C and D*; white arrows).

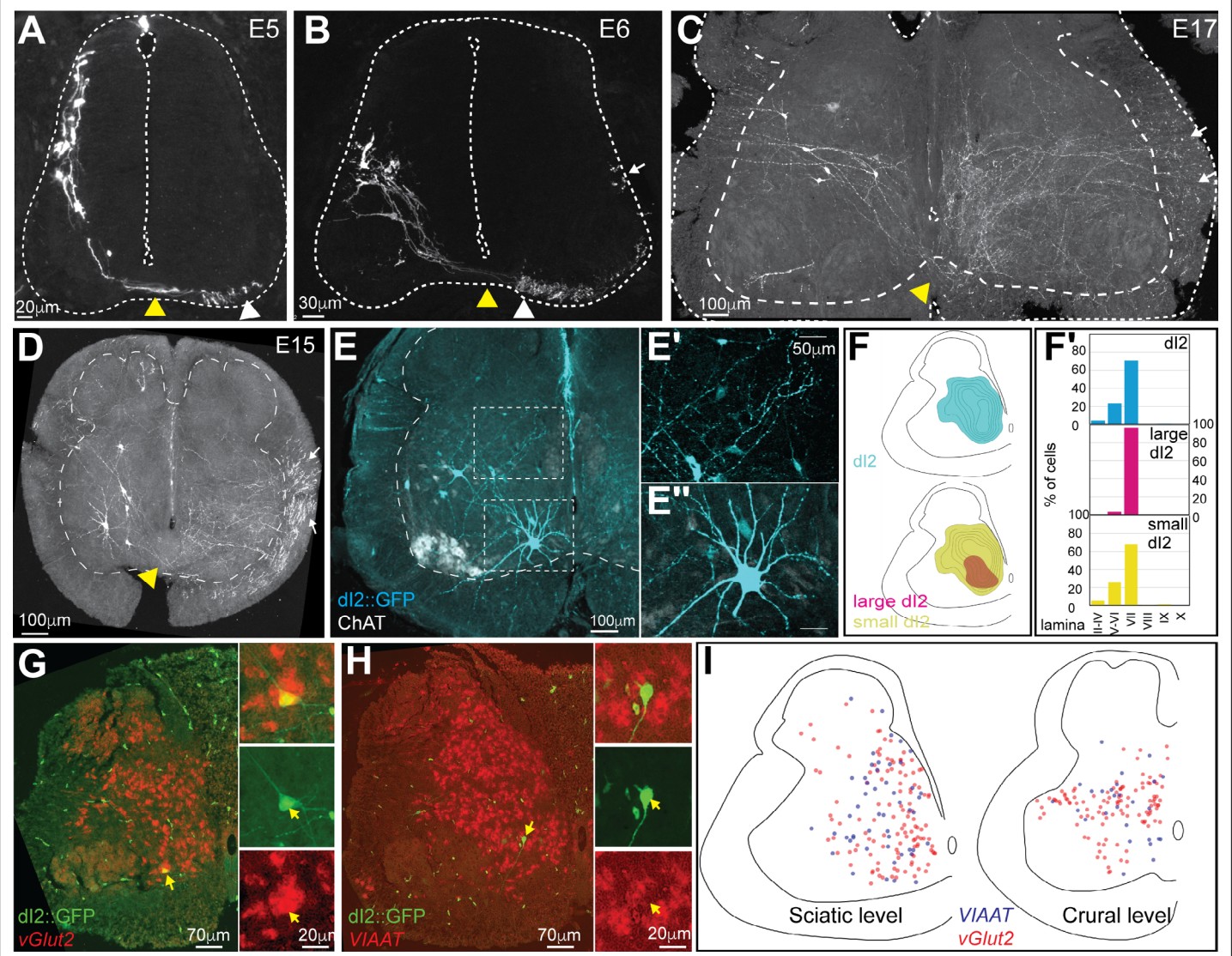

**Figure 1.** Characterization and classification of dI2 neurons during embryonic development. dI2 interneurons (INs) were labeled as cells that expressed both the Foxd3 and Ngn1 enhancers (*Avraham et al., 2009*; see *Figure 1—figure supplement 1A*). (**A–D**) dI2 axonal projection during development. At embryonic day (E) 5 (**A**), postmitotic dI2 neurons assume a dorsolateral position and start to migrate ventrally. At E6 (**B**), dI2 neurons occupy the midlateral domain. At E15-17, dI2 neurons are located at medial lamina VII at the lumbar level (LS3) (**C**) and thoracic level (T1) (**D**). dI2 axons cross the floor plate (yellow arrowheads), turn longitudinally at the ventral funiculus (white arrowheads) and eventually elongate at the lateral funiculus (white arrows). (**E**) Cross-section of an E17 embryo at the lumbar spinal cord (crural plexus level, LS2). Small-diameter dI2 neurons residing in lamina VII (**E'**) and ventromedial large-diameter dI2 neurons in lamina VIII (**E''**). (**F**) Density plots and laminar distribution (**F'**) of dI2 somata at the sciatic plexus level (cyan, N = 374 cells); large-diameter (magenta) and small-diameter dI2 (yellow) INs (N = 33 and N = 344 cells, respectively, from two embryos). (**G, H**) Neurotransmitter phenotype of dI2 neurons. dI2 neurons expressing GFP were subjected to in situ hybridization using the *Vglut2* probe (**G**) or the *VIAAT* probe (**H**). (**I**) Distribution of excitatory (vGlut2, red) and inhibitory (VIAAT, blue) dI2 neurons at the sciatic and crural levels at E17 (N = 172 and 136 neurons, respectively, from two embryos). See *Figure 1—source data 1* and *2*.

The online version of this article includes the following source data and figure supplement(s) for figure 1:

**Source data 1.** Localization of dI2 neurons at the sciatic level.

**Source data 2.** Localization of excitatory and inhibitory dI2 neurons.

**Figure supplement 1.** Targeting, reporters, and activity modifiers used in the study.

**Figure supplement 2.** Differential expression of transcription factor (TF) in dI2 neurons.

**Figure supplement 2—source data 1.** Pattern of expression of transcription factors (TFs) in dI2 neurons.

**Figure supplement 3.** Distribution of dI2 neurons at the embryonic and posthatching spinal cord.

**Figure supplement 3—source data 1.** Localization of dI2 neurons at the crural and brachial levels.

A recent study suggested that dI2 neurons at early stages of development in mice (E9.5–E13.5, comparable to chick E4–8) can be divided into several subclasses based on their genetic signature and degree of maturation (*Delile et al., 2019*). To assess the diversity of dI2 neurons in the chick, the expression of dI2 TFs in dI2::GFP cells was analyzed at E5 before and during ventral migration, at E6 and E14. The early postmitotic dI2::GFP cells at E5 were a homogenous population defined by Foxd3$^+$/Lhx1$^+$/Pou4f1$^+$/Pax2$^-$ (*Figure 1—figure supplement 2A, B, E*). dI2 neurons that underwent ventral migration at E5, as well as at E6 and E14, express variable combinations of Lhx1, Pou4f1, and FoxP1/2/4 (*Figure 1—figure supplement 2C–E*). At E14, approximately 50 % of dI2::GFP cells did not express any of the tested TFs (*Figure 1—figure supplement 2E*), suggesting that the early expression of TFs is required for cell fate acquisition, axon guidance, and target recognition, while their expression is not required after the establishment of the circuitry, as shown for other spinal INs (*Bikoff et al., 2016*). Interestingly, approximately 12 % of ventrally migrating dI2 neurons (from E5 to E14) expressed Pax2 (*Figure 1—figure supplement 2D and E*). Pax2 is associated with a GABAergic inhibitory phenotype (*Cheng et al., 2004*), suggesting that a subpopulation of dI2 are inhibitory neurons. The distribution of excitatory and inhibitory dI2 neurons is also apparent at E17. In situ hybridization on cross-sections of the E17 dI2::GFP-labeled lumbar spinal cord using the vGlut2 probe revealed that 73 % were vGlut2$^+$, while the VIAAT probe, which labels GABAergic and glycinergic inhibitory neurons, measured 27 % VIAAT$^+$ dI2 neurons (*Figure 1G–I*; N = 308 neurons from two embryos). Similar percentages of Gad2- and Slc6a5-dI2-expressing cells were also found in mice (*Delile et al., 2019*). At E13–17 at the caudal lumbar level and at the level of the sciatic plexus, most dI2 neurons are located in the medial aspect of lamina VII. Approximately 91 % of dI2 neurons are small-diameter neurons located in the lateral dorsal aspect of lamina VII, and 9 % are large-diameter neurons. At the lumbar sciatic plexus level, large-diameter dI2 neurons are located mostly at the ventral aspect of lamina VII (*Figure 1F*) and at the level of the crural plexus in the ventral and dorsal aspect of lamina VII (*Figure 1—figure supplement 3A*). The location of the lumbar dI2 neurons, mainly lamina VII, is also apparent in posthatching chicks (P8, *Figure 1—figure supplement 3C* and D). Importantly, large-diameter dI2 neurons were apparent only at the lumbar level (*Figure 1F*, *Figure 1—figure supplement 3A*). The division of large- and small-diameter lumbar dI2 neurons was not reflected in the expression of the tested TFs or in a specific neurotransmitter phenotype; the inhibitory/excitatory ratios were 0.297 ± 0.13 and 0.343 ± 0.02, for large- and small-diameter dI2 neurons, respectively. Hence, dI2 neurons consist of several subpopulations, as has been shown in other spinal INs (*Bikoff et al., 2016*; *Delile et al., 2019*; *Sweeney et al., 2018*).

## Subpopulation of dI2 neurons project to the cerebellum

To study the supraspinal targets of dI2 neurons, axonal and synaptic reporters were expressed in lumbar dI2 neurons (*Figure 2A*). At stage HH18 (E3), dI2 enhancers were co-electroporated with the double conditional axonal reporter membrane-tethered Cherry and the synaptic reporter SV2-GFP (*Figure 1—figure supplement 1A*). Expression in the lumbar spinal cord was attained by using thin electrodes positioned near the lumbar segments. At E17, the stage in which the internal granule layer is formed in the chick cerebellum, the axons and synapses of dI2 neurons were studied. dI2 axons cross the spinal cord at the floor plate at the segmental level, ascend to the cerebellum, enter through the superior cerebellar peduncle, and cross back to the ipsilateral side of the cerebellum (*Figure 2B*). Synaptic boutons were noticeable in the granule layer at the ipsilateral and contralateral sides of the anterior cerebellar lobules (*Figure 2C*). Synaptic boutons were also present in the central cerebellar nuclei (*Figure 2—figure supplement 1A*).

The difference in soma size between dorsally and ventrally located dI2 neurons prompted us to test which dI2 neurons project to the cerebellum. dI2 neurons and precerebellar neurons were colabeled by genetic targeting of dI2 at early stages of embryogenesis (HH18), coupled with intracerebellar injection of replication-defective HSV-LacZ at E15 or PRV-Cherry (*Figure 2A*). Cholera toxin subunit B (CTB) was coinjected with PRV-Cherry to verify primary infection of precerebellar neurons. Spinal neurons retrogradely labeled from the cerebellum consist of double-crossed VSCT neurons and ipsilaterally projecting DSCT neurons. However, dI2 neurons, double labeled by genetic targeting and retrograde labeling from the cerebellum, are all VSCT neurons since dI2 neurons are commissural neurons. The soma distributions of precerebellar neurons, dI2 neurons, and dI2 synapses overlapped at the sciatic level (*Figure 2E, G and H*) and to a lesser extent at the crural level (*Figure 2—figure*

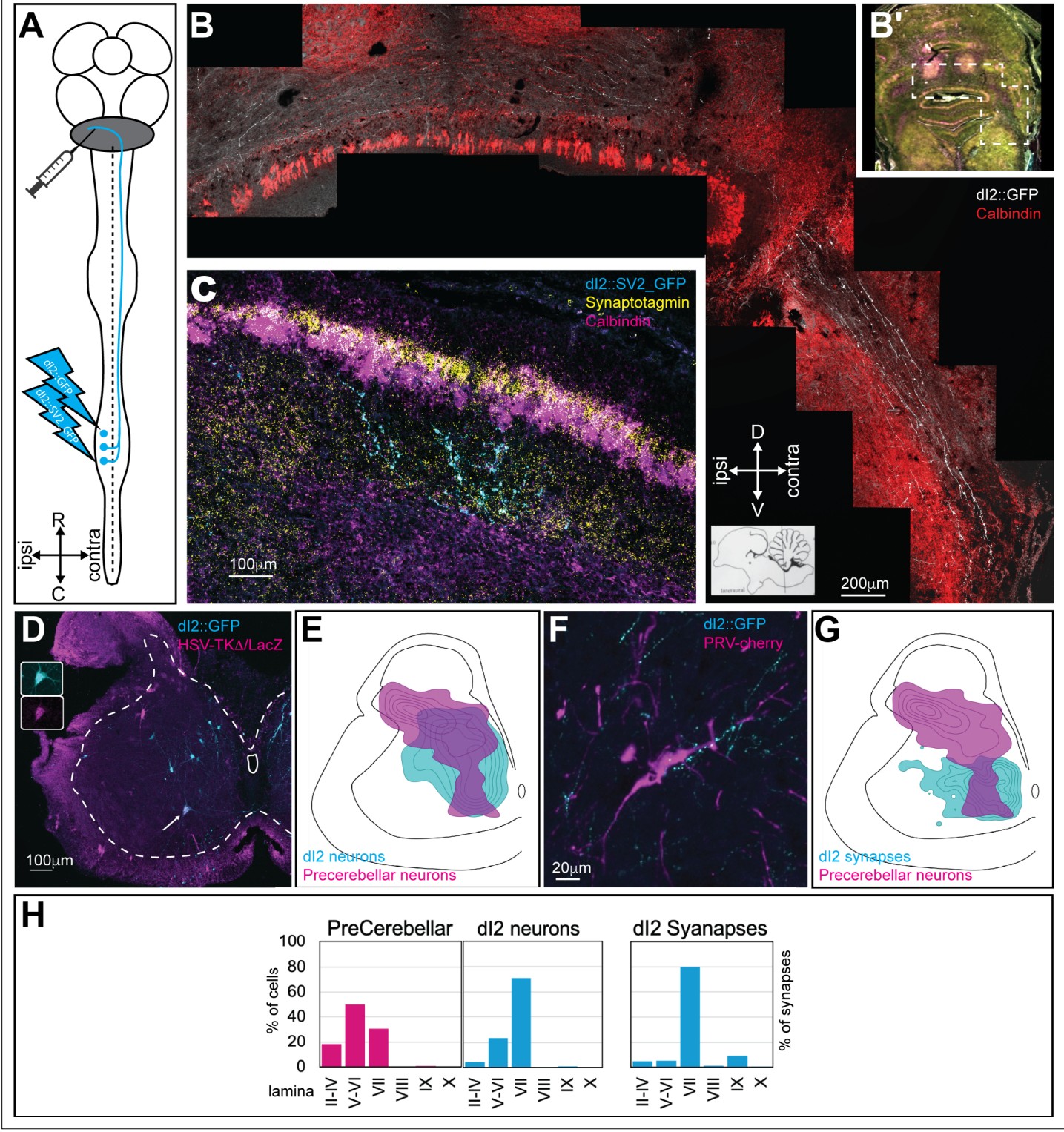

**Figure 2.** dI2 neurons project to the cerebellum. (**A**) Experimental setup for labeling dI2 neurons that project to the cerebellum. dI2 neurons were genetically targeted at HH18, and precerebellar neurons were labeled using intracerebellar injection of replication-defective HSV-LacZ or PRV-Cherry at embryonic day (E) 15. The abbreviations in the coordinates: R: rostral; C: caudal. (**B**) A cross-section of E17 brainstem and cerebellum. The dashed polygon in (**B'**) is magnified in (**B**). dI2 axons reach the cerebellum, enter into it via the superior cerebellar peduncle, and cross the cerebellar midline. Calbindin (Purkinje neurons, magenta [**B'**] or red [**B**]). Abbreviations in the coordinates: D: dorsal; V: ventral. (**C**) A cross-section of E17 cerebellar cortex. Lumbar-originating dI2 synapses (cyan) in the granular layer of the anterior cerebellar cortex. Calbindin (Purkinje neurons, magenta), synaptotagmin

*Figure 2 continued on next page*

*Figure 2 continued*

(yellow). (**D**) A cross-section of an E15 embryo at the lumbar spinal cord level (sciatic plexus level). Precerebellar neurons were infected and labeled with HSV-LacZ (magenta), and dI2 neurons expressed GFP (cyan). A large-diameter dI2 neuron coexpressing LacZ and GFP is indicated by an arrow (magnification of the two channels in the insets). (**E**) Density plots of dI2 and precerebellar neurons (density values 10–90%) in the sciatic plexus segments (N = 374 and N = 289 cells, respectively). (**F**) PRV-Cherry-labeled precerebellar neurons (magenta) are in contact with dI2 axonal terminals (cyan). (**G**) Density plots of dI2 synapses and precerebellar neuron somata (density values 10–90%) in the sciatic plexus segments (N = 4735 synapses and N = 289 cells, respectively). (**H**) The laminar distribution of precerebellar neurons, dI2 neurons, and dI2 synapses at the sciatic level. See *Figure 2—source data 1*.

The online version of this article includes the following source data and figure supplement(s) for figure 2:

**Source data 1.** Localization of precerebellar neurons and dI2 synapses at the sciatic level.

**Figure supplement 1.** Cerebellar and central cerebellar nucleus targets of dI2 neurons.

**Figure supplement 1—source data 1.** Localization of precerebellar neurons and dI2 synapses at the crural level.

*supplement 1B–D*). We found that large-diameter dI2 neurons were mostly colabeled, most of them in the ventral aspect of lamina VII (*Figure 2D*). Interestingly, many of the cherry⁺ and LacZ⁺ neurons were contacted by dI2 axons (*Figure 2F*), suggesting that dI2 neurons innervate precerebellar neurons.

Segmental crossing at the lumbar level and recrossing back to the ipsilateral side at the cerebellum are characteristic of the VSCT projection pattern. To measure the proportion of precerebellar neurons in dI2, we used sparse labeling of lumbar dI2 neurons coupled with whole-mount imaging of the E13 spinal cord from the sacral level to the cerebellum utilizing light-sheet microscopy and iDISCO imaging (*Belle et al., 2014*; *Renier et al., 2014*; *Figure 3*; *Video 1*, *Video 2*, *Video 3*, *Video 4*). 85 neurons were labeled at the lumbar level, 17 of which were large-diameter neurons. The axons of all dI2 neurons cross the midline. Longitudinally projecting axons were apparent at the contralateral VF and LF (*Figure 3A–F*; *Video 1*, *Video 2*, *Video 3*). In the gray matter, axonal ramifications originating from collateral branching were apparent along the entire extent of the spinal cord (*Figure 3A–C*; *Video 1*, *Video 2*, *Video 3*). Bifurcation of the longitudinal axons was also apparent. The bifurcating branch elongated rostrally for a few segments and subsequently turned transversely into the spinal cord (*Figure 3—figure supplement 1A*). We counted the numbers of longitudinally projecting axons at different levels (*Figure 3A and D-H*): 22 axons at LS1 (*Figure 3D*), 30 at T1 (*Figure 3E*), 21 at C9 (*Figure 3F*), 8 at the rostral brain stem (*Figure 3G*), and 7 entering the cerebellum via the superior cerebellar peduncle (*Figure 3H*). The elevated number of axons in T1 levels likely reflects longitudinal bifurcations. Considering that the spinal cord was observed at E13, a relatively early stage of development, it is likely that additional axons enter the cerebellum at later stages of development.

To evaluate the complexity of the branching pattern, we reconstructed the axonal projection and branching pattern of two large-diameter dI2 neurons at the lumbar and brachial levels (*Figure 3B and C*). Numerous collaterals that penetrate the spinal cord along its entire length were evident. Importantly, VSCT dI2 neurons projected to spinal targets at the lumbar, thoracic, and brachial spinal levels and to the brain stem and cerebellum (*Figure 3*).

To measure the proportion of dI2 in VSCT neurons, we labeled VSCT axons with GFP and dI2 axons with Cherry (for experimental design, see *Figure 3—figure supplement 1B and C*). The number of axons expressing the reporters at the contralateral superior cerebellar peduncle was scored. 10% of the VSCT axons belonged to dI2 neurons (*Figure 3—figure supplement 1D*). Thus, the large-diameter dI2 neurons constitute 10 % of the VSCT neurons, consistent with the anatomical observation that the VSCT comprises a heterogeneous population of INs (*Jankowska and Hammar, 2013*; *Stecina et al., 2013*).

## Mapping the synaptic input and output of dI2 neurons

To obtain the connectome of dI2 neurons, we employed enhancer-mediated synaptic labeling of presynaptic neurons coupled with soma labeling of postsynaptic neurons. We used three criteria for assessing synaptic contact: (1) the likelihood of connectivity was examined by spatial overlap of axonal terminals from the presumed presynaptic neurons and the somata of the postsynaptic neurons; (2) synaptic boutons were detected on the somatodendritic membrane of postsynaptic neuron; and (3) colabeling was observed between the presynaptic reporter and synaptotagmin (syn). We used confocal imaging and 3D reconstitution to score overlap (*Figure 4—figure supplement 1A, B*).

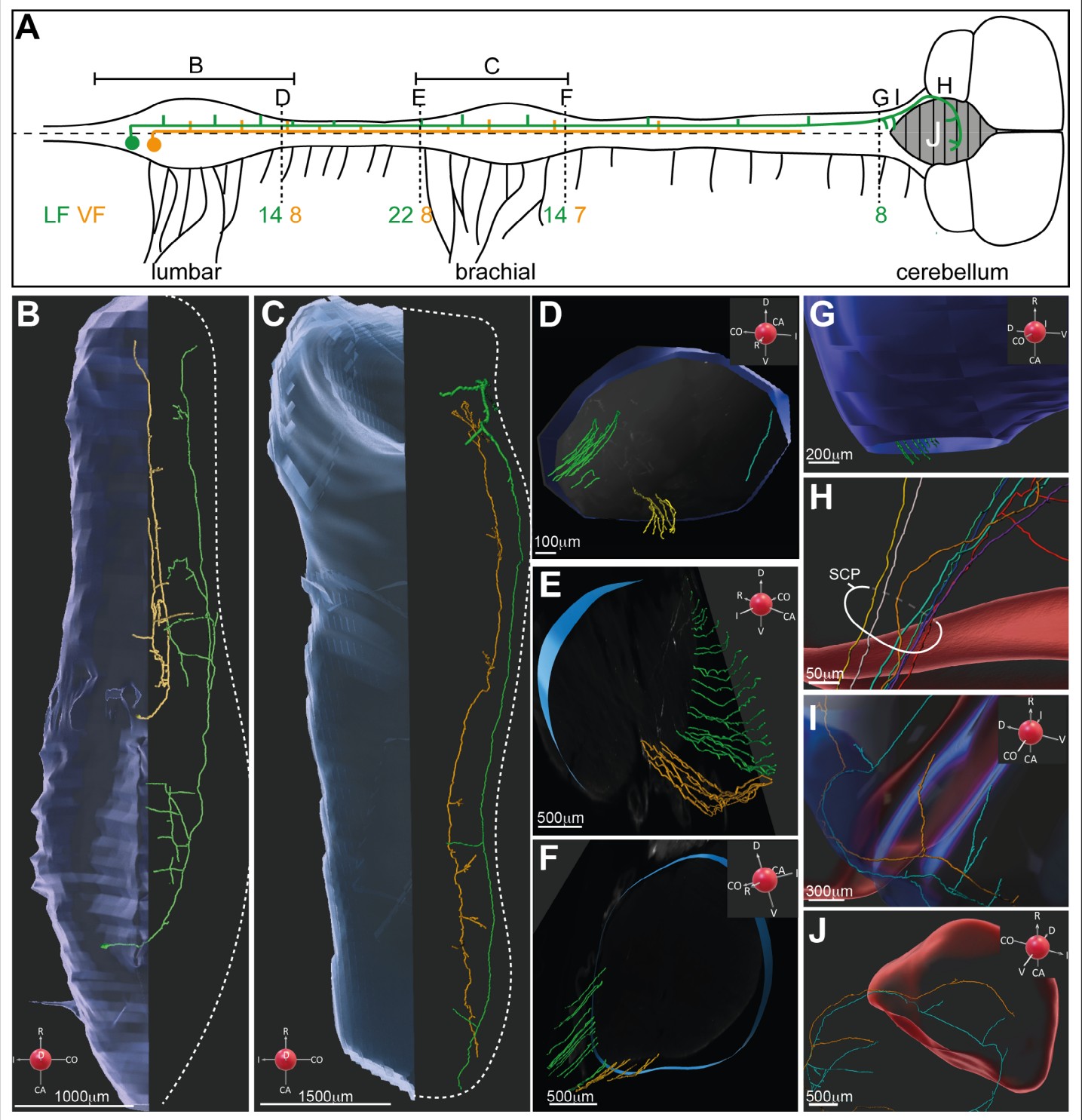

**Figure 3.** 3D reconstruction of dI2 neurons along the rostrocaudal axis. (**A**) Spinal cord scheme describing dI2 axonal projection along the rostrocaudal axis (caudal is to the left, and rostral is to the right). The full lines represent the lumbar and brachial levels shown in (**B**) and (**C**). The broken lines represent the cross-sections shown in (**D–G**). The number of axons (and the funicular division) along the rostrocaudal axis is indicated adjacent to the corresponding letters (**D–G**). (**B, C**) Two representative dI2 neurons projecting their axons in the lateral funiculus (LF; green) and ventral funiculus (VF; yellow) at the lumbar (**B**) and brachial (**C**) levels. Numerous axonal collaterals are apparent. (**D–F**) Cross-sections at different levels of the spinal cord showing dI2 axons exiting the rostral end of the lumbar segments (**D**), entering the caudal brachial level (**E**), and exiting the rostral brachial level (**F**). Green: LF on the contralateral side (cLF); orange: VF on the contralateral side (cVF); cyan: LF on the ipsilateral side (iLF). (**G–J**) dI2 axons in the brainstem and cerebellum. (**G**) Axons entering the brainstem are indicated in green. (**H**) dI2 axons enter the cerebellum via the superior cerebellar peduncle

*Figure 3 continued on next page*

*Figure 3 continued*

(SCP). (**I**) Collaterals projecting into the brainstem. (**J**) The axons cross the cerebellar midline back to the ipsilateral side (two representative axons). A coordinate system is supplied in (**B–G**, **I**, **J**).

The online version of this article includes the following source data and figure supplement(s) for figure 3:

**Figure supplement 1.** dI2 projection neurons constitute 10 % of neurons in the ventral spinocerebellar tract (VSCT).

**Figure supplement 1—source data 1.** Analysis if the dI2 and ventral spinocerebellar tract (VSCT) axons at the superior peduncle.

As a proof of concept, we tested the colabeling of dI2::SV2-GFP and syn in dI2 to contralateral pre-MN synapses. Of 144 genetically labeled boutons, 121 (84%) were syn⁺. The syn⁻ boutons were significantly smaller (*Figure 4—figure supplement 1C*). We set a volume threshold (0.07 µm³, *Figure 4—figure supplement 1C*), and small-volume SV2-GFP boutons were not considered synapses in the study. Using these criteria, we mapped the putative pre-dI2 and post-dI2 neurons.

## dI2 neurons receive synaptic input from pre-MNs and sensory neurons

To assess the synaptic input to dI2 neurons, we investigated their synaptic connectivity with the following: (1) dorsal root ganglion (DRG) neurons (*Figure 4A*). (2) Ipsilateral pre-MNs. General ipsilateral pre-MNs were labeled by injecting a PRV-Cherry virus into the ipsilateral hindlimb musculature (*Hadas et al., 2014*; *Figure 4B*). Two genetically defined classes of pre-MNs were examined: dI1i excitatory INs (*Figure 4C*, *Figure 4—figure supplement 2*) and the V1 inhibitory pre-MN population (*Bikoff et al., 2016*; *Gosgnach et al., 2006*; *Figure 4D*). (3) Reticulospinal tract neurons (*Figure 4E*). dI2, DRG, V1, and dI1 neurons were labeled using specific enhancers (*Figure 1—figure supplement 1A*).

A density profile of the axons of DRG neurons (*Figure 4A*, *Figure 4—figure supplement 3A–D*) was aligned with the density plots of the dI2 somata. Overlap between the axonal terminals of DRG neurons was evident (*Figure 4A*, *Figure 4—figure supplement 3C and D*). Contact between DRG axons and dI2 neurons was mainly apparent in the dorsal dI2 neurons, while the ventral dI2 neurons received little to no input from DRG neurons (2.8 ± 2.4 vs. 16.9 ± 11.3 contacts per neuron for ventral and dorsal dI2 neurons, respectively; p<1e-5; *Figure 4A*, *Figure 4—figure supplement 3E*). In contrast, large- and small-diameter dI2 neurons did not exhibit a significant difference in DRG axon contacts (10.4 ± 14.9 vs. 7.8 ± 5.7 contacts per neuron for large and small dI2 neurons, respectively; p=0.4) (*Figure 4—figure supplement 3F*).

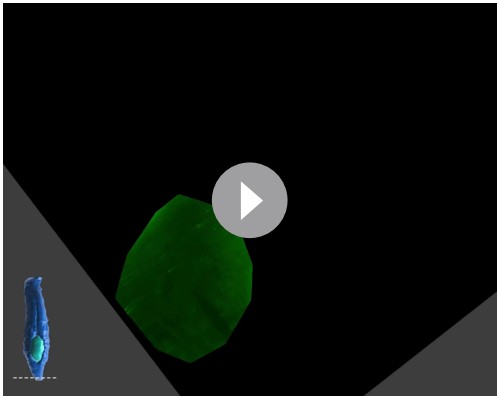

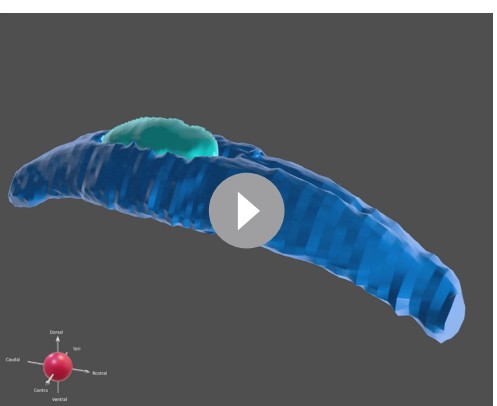

**Video 1.** dI2 interneurons (INs): transverse sections of lumbar segments. Caudal to rostral transverse images of light-sheet microscopy images along the lumbar segments of an embryonic day (E) 13 spinal cord expressing Cherry in dI2 neurons (a reference to the location of the section is shown at the bottom left). dI2 cell bodies and axons are visible. Examples of large- and small-diameter dI2 neurons are indicated by arrows. The concentration of axons on the side contralateral to electroporation is clear.

https://elifesciences.org/articles/62001/figures#video1

**Video 2.** dI2 interneurons (INs): lumbar segments. 3D reconstruction of light-sheet microscopy images of the lumbar spinal cord. In the transparent mode, dI2 axons are apparent on the contralateral side. The trajectory of two representative axons (ventral and lateral projection axons in yellow and green, respectively) was reconstructed. Significant branching is apparent. All the axons exiting the lumbar segments are visible. A coordinate system is supplied in key frames.

https://elifesciences.org/articles/62001/figures#video2

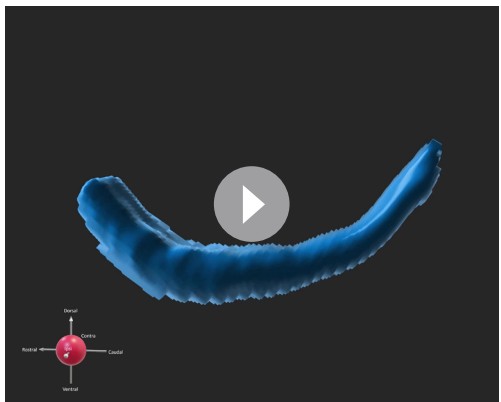

**Video 3.** dI2 interneurons (INs): brachial segments. 3D reconstruction of light-sheet microscopy images of dI2 axons entering and exiting the brachial spinal cord. Two representative axons (ventral and lateral projection axons in yellow and green, respectively) were followed, and their collaterals along the spinal cord are demonstrated. A coordinate system is supplied in key frames.

https://elifesciences.org/articles/62001/figures#video3

The density plot of PRV-labeled pre-MNs overlapped with the density plots of the dI2 somata (*Figure 4B*), and the axonal terminals of pre-MNs were visible on dI2 somata and dendrites (*Figure 4B',B''*), suggesting that pre-MNs contacted dI2 neurons. To solidify the evidence for pre-MN/dI2 connectivity, we used synaptic reporters expressed in genetically identified dI1 and V1 pre-MNs. Excitatory dI1 synapses and inhibitory V1 synapses overlapped with the density plots of the dI2 somata (*Figure 4C and D*). Synaptic connections, evaluated by boutons found on dI2 dendrites and somata, were apparent from V1 and dI1i (*Figure 4B–D*, *Figure 4—figure supplement 3G*).

Serotonergic neurons are the main reticulospinal input to VSCT in cat (*Hammar et al., 2004*; *Hammar and Maxwell, 2002*). Serotonergic synapses were concentrated on motor neurons and were not observed on dI2 neurons (*Figure 4E*, *Figure 4—figure supplement 3H*). Double labeling of 5-HT and dI2 neurons did not reveal any synaptic input. The lack of synaptic serotonergic input may be related to the difference in species or may suggest that other, non-dI2 VSCT neurons located adjacent to motoneurons are contacted by the reticulospinal neurons. The analysis of synaptic inputs supports the concept that dI2 neurons constitute part of the VSCT. These cells receive input from sensory afferents and inhibitory and excitatory pre-MNs and project to the cerebellum.

## dI2 neurons innervate contralateral lumbar and brachial pre-MNs and dI2 neurons

Axon collaterals of dI2 invade the gray matter along the entire length of the spinal cord, as revealed by whole-mount staining of spinal cords electroporated with an alkaline phosphatase reporter (dI2::AP) (*Figure 5A*), cross-sections of dI2 neurons expressing membrane-tethered EGFP (*Figure 5B*), and light-sheet microscopy analysis (*Figure 3—figure supplement 1*; *Video 1*, *Video 2*, *Video 3*). The region innervated by dI2 collaterals (arrow in *Figure 5B*) overlaps with that of the V0 and V1 pre-MNs (*Lai et al., 2016*; *Griener et al., 2015*) as well as with that of the contralateral dI2 neurons (*Figures 1 and 5B*). To assess the potential spinal targets of dI2 neurons, we inspected the degree of overlap between dI2 synapses and dI2 somata (*Figure 5C*), ipsilateral pre-MNs (*Figure 5D*), and contralateral pre-MNs (*Figure 5E*). The alignment revealed an overlap of dI2 synapses with ipsilateral/contralateral pre-MNs and dI2 neurons (*Figure 5C–E*), supporting their potential connectivity. Labeling of dI2 synapses coupled with labeling of the above neuronal population showed dI2 synaptic boutons on pre-MNs and dI2 neurons at the lumbar level (*Figure 5C–E*, *Figure 5—figure supplement 1A–C*).

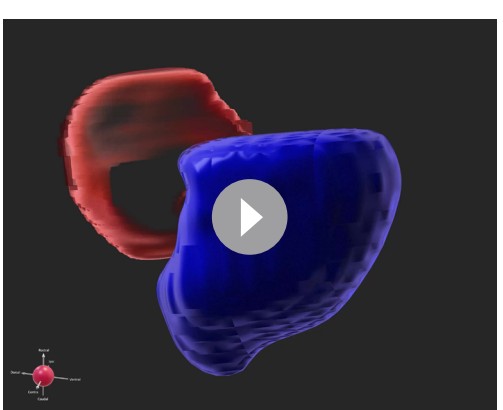

**Video 4.** dI2 interneurons (INs): brainstem and cerebellum. 3D reconstruction of light-sheet microscopy images of dI2 axons projecting into the brain stem and the cerebellum (blue and red, respectively). Cerebellum midline crossing is demonstrated for two representative axons. The axonal collaterals to the brainstem are apparent. A coordinate system is supplied in key frames.

https://elifesciences.org/articles/62001/figures#video4

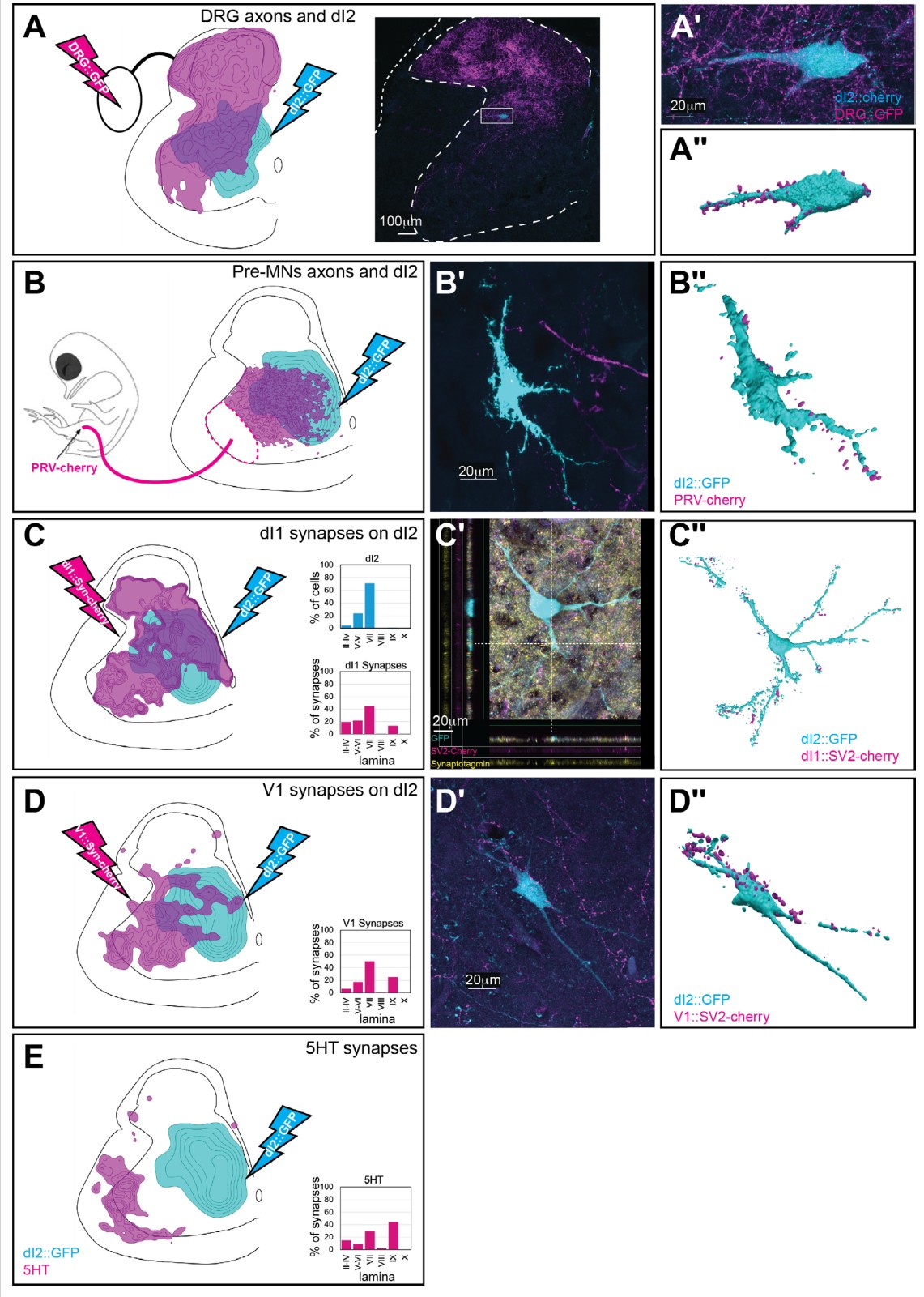

**Figure 4.** Synaptic inputs to dI2 neurons. Schematic representations of the experimental design for labeling dI2::GFP or dI2::Cherry interneurons (INs; cyan) and potential sources of synaptic inputs (magenta). The soma densities of dI2 INs and the synaptic densities are illustrated in (**A–E**). The density values presented are 10–80%, 20–80%, 25–80%, 30–50%, and 20–80%, respectively. The laminar distributions are illustrated on the right side of (**A–E**). Examples of dI2 neurons contacted by axons or synaptic boutons are shown in (**A'–D'**), and their 3D reconstruction is shown in (**A"–D"**). Genetic

*Figure 4 continued on next page*

*Figure 4 continued*

labeling was achieved using specific enhancers (*Figure 1—figure supplement 1A*) introduced by electroporation at HH18. (**A**) Dorsal root ganglion (DRG) neurons form contacts on dI2 neurons. Inset in (**A**): cross-section of embryonic day (E) 17 embryos at the crural plexus level of the lumbar cord. A dorsally located dI2 neuron contacted by numerous sensory afferents, magnified in (**A′**) and 3D-reconstructed in (**A″**) (N = 18 sections, the scheme was constructed based on one representative embryo). (**B**) Premotor neurons (pre-MNs) form contacts on dI2 neurons. dI2 neurons were labeled at HH18. At E13, PRV virus was injected into the leg musculature, and the embryo was incubated until the infection of the pre-MNs (39 hr) (N = 34 sections, the scheme was constructed based on two representative embryos). (**C**) dI1 neurons form synapses on dI2 neurons. (N = 8568 synapses, 2 embryos). (**C′**) A representative SV2::Cherry synapse on dI2 dendrites positive for synaptotagmin. Demonstrated by horizontal and vertical optical sections in the Z-axis (see cursors and color channels). (**D**) V1 neurons form synapses on dI2 neurons (N = 1923 synapses, 2 embryos). (**E**) dI2 neurons are not contacted by 5-HT synaptic terminals (N = 1718 synapses, 1 embryo). E17 cross-sections of dI2::GFP-labeled embryos were stained for 5-HT. See *Figure 4—source data 1*.

The online version of this article includes the following source data and figure supplement(s) for figure 4:

**Source data 1.** Localizations of pre-dI2 terminals and synapses at the sciatic level.

**Figure supplement 1.** Validation of the synaptic reporter as an indicator of synapses.

**Figure supplement 1—source data 1.** Validation of the use of SV2-GFP reporter as an indicator for synapses.

**Figure supplement 2.** dI1i neurons are premotor neurons (pre-MNs).

**Figure supplement 2—source data 1.** Localization of dI1 synapses.

**Figure supplement 3.** Input of dorsal root ganglion (DRG), dI1, and 5-HT neurons to dI2 neurons at the level of the crural plexus.

**Figure supplement 3—source data 1.** Distribution of dorsal root ganglion (DRG) terminals on dI2 neurons.

**Figure supplement 3—source data 2.** Localizations of pre-dI2 terminals and synapses at the crural level.

---

The pattern of dI2 collaterals along the entire rostrocaudal axis (*Figure 3A–C*, *Figure 5A*) suggests that dI2 neurons innervate contralateral pre-MNs and dI2 neurons at multiple levels. To test this hypothesis, labeling of lumbar dI2 neurons was coupled with labeling of brachial pre-MNs and dI2 somata by injecting PRV into the wing musculature or electroporating a reporter into brachial dI2 neurons, respectively (*Figure 5F*, *Figure 5—figure supplement 1D*). dI2 synapses overlapped with the putative targets, and synaptic boutons originating from lumbar-level dI2 neurons were apparent on dI2 neurons and on the contralateral and ipsilateral pre-MNs of the wings (*Figure 5F*, *Figure 5—figure supplement 1D and E*).

Neuronal and synaptic labeling experiments showed that lumbar dI2 neurons innervate the cerebellum, lumbar and brachial pre-MNs, and contralateral dI2 neurons. Hence, dI2 neurons may relay peripheral and intraspinal information to the cerebellum and to the contralateral lumbar and brachial motor control centers.

## Silencing of dI2 neurons impairs the stability of bipedal stepping

The synaptic input to dI2 neurons and their putative targets implicates them as relaying information about motor activity to the contralateral spinal cord and the cerebellum. Thus, we hypothesized that manipulation of their neuronal activity may affect the dynamic profile of stepping. To study the physiological role of dI2 neurons, we silenced their activity by expressing the tetanus toxin (*TeTX*) light chain gene, which blocks synaptic transmission (*Yamamoto et al., 2003*), in the bilateral lumbar dI2 neurons. EGFP was cotargeted in a 2/1 TeTX/EGFP ratio for post hoc analysis of the efficacy of electroporation (*Supplementary file 1*). Chicks expressing EGFP in dI2 neurons and chicks that did not undergo electroporation were used as controls. To maximize the number of targeted dI2 neurons, we combined genetic targeting with the *Foxd3* enhancer and spatial placement of the electrodes at the dorsal lumbar spinal cord (*Figure 1—figure supplement 1A*). Embryos were electroporated at HH18. Upon hatching, chicks were trained for targeted overground locomotion. The gait parameters of four controls and 5 *TeTX*-treated chicks were measured while chicks were walking toward their imprinting trainer along a horizontal track (6–20 walking sessions, 5–8 strides each, per chick).

To test whether silencing of dI2 neurons impairs posthatching development and muscle strength, the chicks were weighed at P8, and their foot grip strength was evaluated on the same day. All chicks were of comparable weight (average – 144.7 ± 12.1 g; *Supplementary file 1*). As a functional measure of foot grip, we tested the ability of the chicks to maintain balance on a tilted mesh surface. *TeTX*-manipulated chicks and control chicks maintained balance on the tilted surface up to 63–70°,

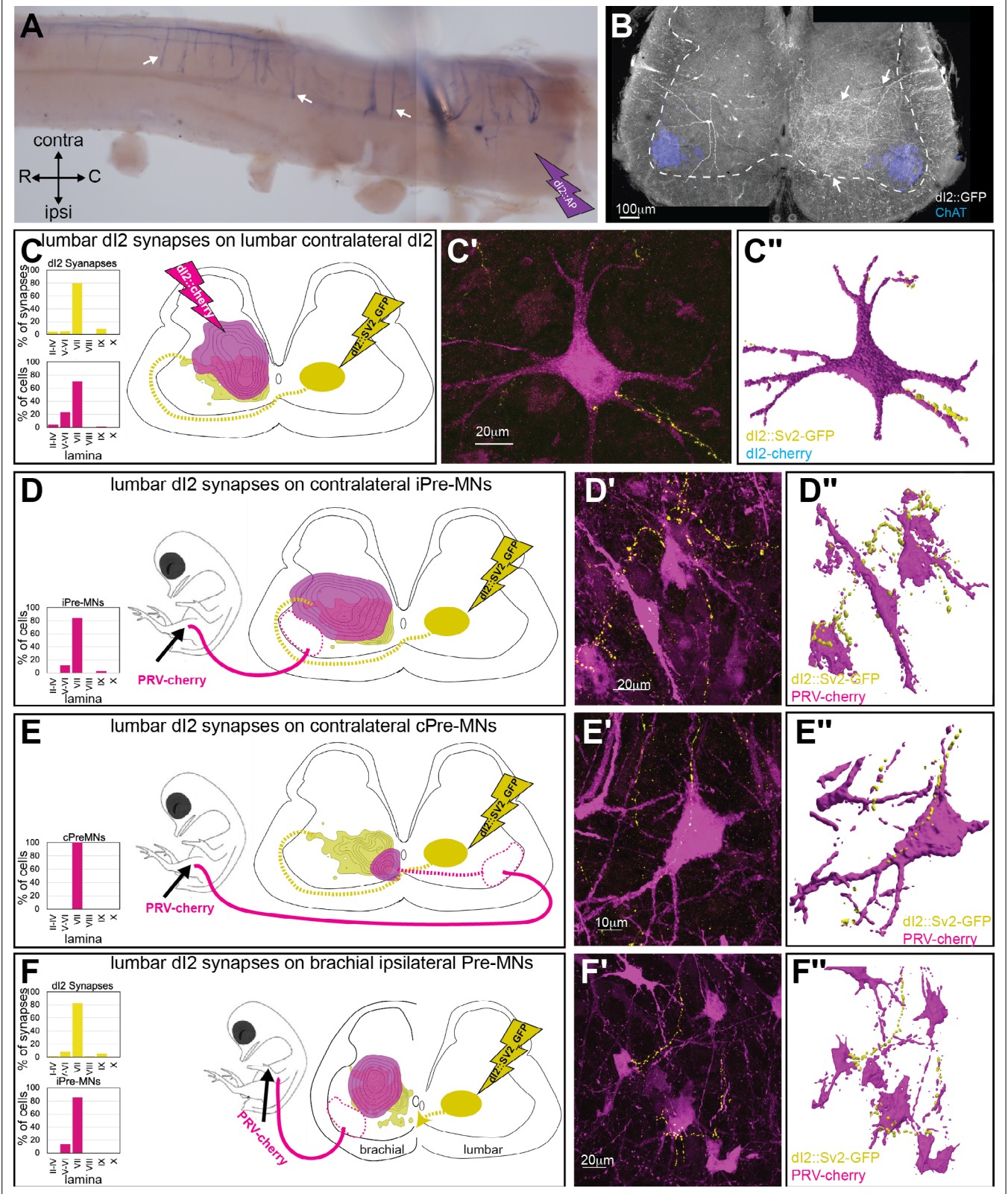

**Figure 5.** Spinal synaptic targets of dI2 neurons. (**A**) Whole-mount staining of the spinal cord (thoracic segments) expressing alkaline phosphatase (AP) in dI2 neurons. The lumbar dI2 neurons (not included in the image) were labeled with AP. dI2 axon collaterals project and into the spinal cord (arrows). Abbreviations in the coordinates: rostral: R: caudal: C. (**B**) Cross-section of an embryonic day (E) 17 embryo at the crural plexus level of the lumbar spinal cord. Axonal collaterals (white arrow) penetrating the gray matter of the contralateral side are evident. Schematic representations of the experimental

*Figure 5 continued on next page*

Figure 5 continued

design for labeling synapses (dl2::SV2-GFP, yellow) and potential targets (magenta) supplemented by cell soma density and dl2 synaptic densities are illustrated in (C–F). The laminar distribution of the somata and synapses is illustrated on the right side of (C–F). Examples of target neurons contacting synaptic boutons of dl2 neurons are shown in (C'–F'), and their 3D reconstruction is shown in (C"–F"). Genetic labeling was achieved using dl2 enhancers (*Figure 1—figure supplement 1A*) electroporated at HH18. Premotor neurons (pre-MNs) were labeled by injection of PRV-Cherry into the hindlimbs (D, E) or the forelimb (F) musculature at E13. The embryos were incubated until the pre-MNs were infected (39 hr). (C) dl2 neurons innervate contralateral dl2 neurons (N = 4735 synapses and N = 374 cells, respectively, two embryos). (D) dl2 neurons innervate ipsilateral projections of pre-MNs at the sciatic plexus level (N = 4735 synapses and N = 936 cells, respectively, scheme was done based on one representative embryo). (E) dl2 neurons innervate contralaterally projecting pre-MNs at the sciatic plexus level (N = 4735 synapses and N = 47 cells, respectively, scheme was done based on one representative embryo). (F) dl2 neurons innervate ipsilaterally projecting pre-MNs at the brachial level (N = 2215 synapses and N = 286 cells, respectively, three embryos). See *Figure 5—source data 1*.

The online version of this article includes the following figure supplement(s) for figure 5:

**Source data 1.** Localization of dl2 synapses on post-dl2 neurons at the sciatic and brachial levels.

**Figure supplement 1.** Spinal targets of dl2 at the crural and brachial levels.

with no apparent statistically significant differences (*Supplementary file 1*). Thus, manipulation of dl2 neuronal activity did not impair the development or balance or muscle strength.

Analysis of overground locomotion in the control and *TeTX*-treated chicks revealed no significant differences in swing velocity or striding pattern. A 180° out-of-phase pattern was found during stepping in all the manipulated and control chicks (*Figure 6—figure supplement 1A*, *Table 1*). However, substantial differences were scored in stability parameters: *TeTX* chicks exhibited whole-body collapses during stepping (*Figures 6B and C and 7A*), a wide-base gait (*Table 2*), and variable limb movements (*Figure 6A, D and E*; *Figure 7B and C*; *Figure 7—figure supplement 1*; *Table 3*).

## Whole-body collapses

A collapse was scored as a decline in knee height below 85 % of the average knee height at the stance phase of the step (arrow in *Figure 6C*). We measured the number of collapses in 50–190 steps. In control chicks, collapses occurred in 0.53% ± 0.92% of the steps. In *TeTX*-manipulated chicks, we observed collapses in 19.46% ± 8.3% of the steps, which was significantly different from the rate in controls (*Figure 7A*). As some collapses were followed by an overextension ('overshoot' in leg elevation), also manifested in the profile of the knee height trajectory during the swing phase (*Figure 6D*), we further studied the relationship between the two phenomena. In general, there was high variability between chicks in this aspect.

Most collapses (64.9% ± 19.7%) were not preceded or followed by overextension. About 22 % of the collapses (22.47% ± 21.5% of collapses, e.g., arrowhead in *Figure 6C*) were followed by

**Table 1.** Stride velocity and left-right phase in control and tetanus toxin (TeTX-manipulated chicks). Swing velocity and left-right phase were measured and calculated as described in Materials and methods. The Watson–Williams test of the phase data (circular ANOVA) was not statistically significant.

| Chick | Mean swing velocity (cm/s) | Mean left-right phase (°) | # of steps |
|---|---|---|---|
| TeTX1 | 46.78 ± 22.13 | 184.679 ± 33.003 | 113 |
| TeTX2 | 62.24 ± 20.17 | 182.293 ± 32.01 | 63 |
| TeTX3 | 48.06 ± 20.04 | 180.784 ± 31.064 | 69 |
| TeTX4 | 57.24 ± 24.35 | 180.502 ± 36.291 | 59 |
| TeTX5 | 36.66 ± 17.61 | 181.97 ± 35.787 | 93 |
| Control 1 (GFP) | 79.65 ± 37.77 | 182.369 ± 35.366 | 47 |
| Control 2 (GFP) | 41.91 ± 20.41 | 182.384 ± 26.708 | 19 |
| Control 3 (not electroporated) | 41.09 ± 16.59 | N.D. | 121 |
| Control 4 (not electroporated) | 42.3 ± 30.91 | N.D. | 51 |

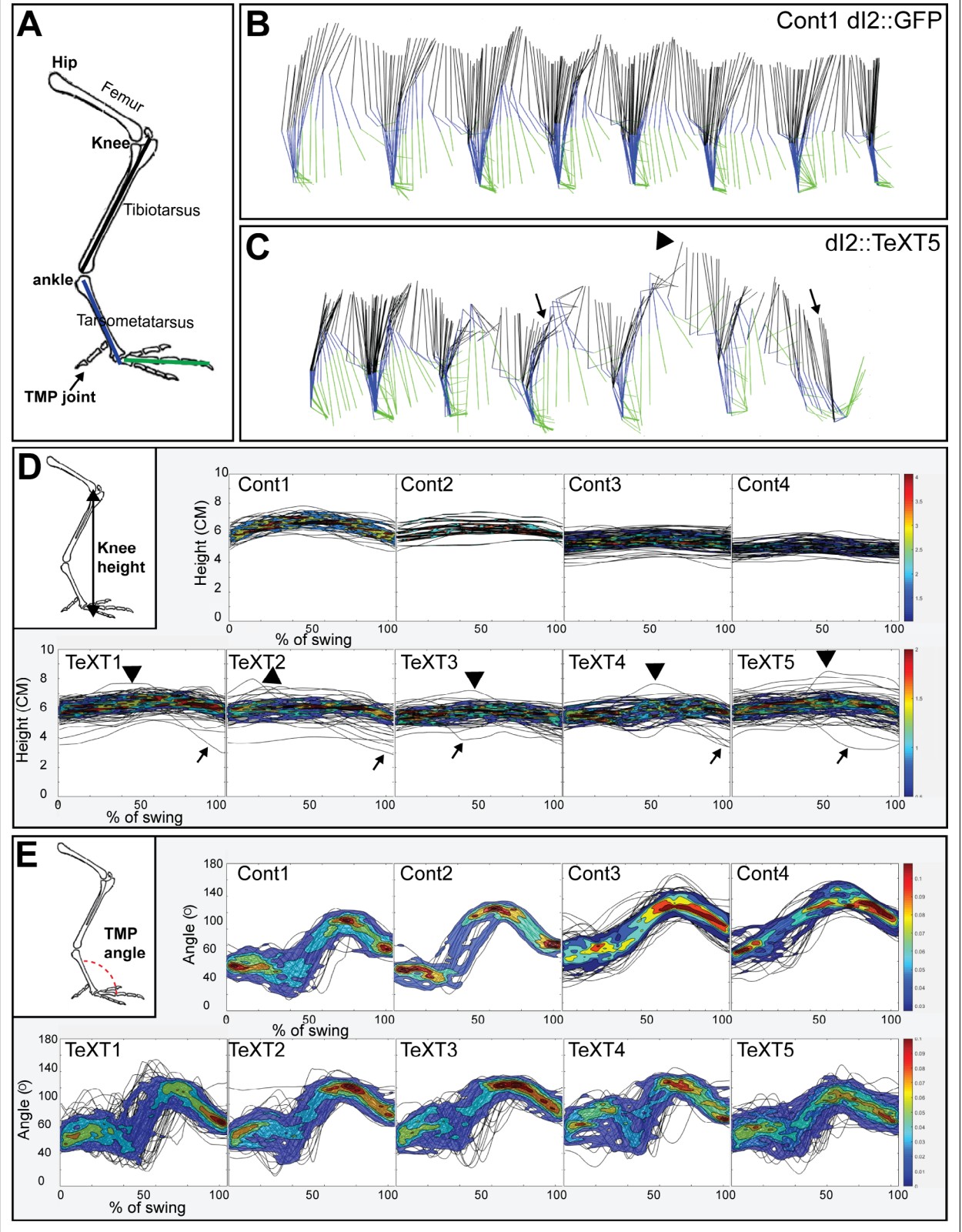

**Figure 6.** Kinematic analysis of locomotion in posthatching chicks following the silencing of dI2 neurons. (**A**) Schematic illustration of chick hindlimb joints (bold) and bones (regular). The knee joint connects the femur and the tibiotarsus, and the ankle connects the tibiotarsus and the tarsometatarsus, which connects to the phalanges at the TMP joint. During the swing phase of birds, ankle flexion leads to foot elevation, while the knee is relatively stable. (**B, C**) Stick diagrams of stepping in a control chicken d2::GFP (**B**) and in a d2::TeTX chicken (**C**). Arrows indicate collapses, and overshoots

*Figure 6 continued*

are denoted by arrowheads. (**D**) Overlays of knee height (demonstrated in insert) trajectories during the swing phase in all analyzed steps of each of the control and *TeTX*-treated posthatching day (P) 8 hatchlings are shown superimposed with the respective 20–80% color-coded density plots as a function of the percentage of swing (see text and Materials and methods). Arrows indicate collapses, and overshoots are indicated by arrowheads. (**E**) The angular trajectories of the TMP joint (shown in insert) during the swing phase in all analyzed strides of each of the control and *TeTX*-treated P8 hatchlings are shown superimposed on the respective 20–80% color coded density plots as a function of the percentage of swing (see text and Materials and methods). See *Figure 6—source data 1*, *Figure 6—source data 2*, *Figure 6—source data 3*.

The online version of this article includes the following source data and figure supplement(s) for figure 6:

**Source data 1.** Analysis of knee height trajectories during the swing phase.

**Source data 2.** Analysis of TMP angles during the swing phase.

**Source data 3.** Statistical analysis of knee height trajectories and TMP angles.

**Figure supplement 1.** Locomotion characteristics of control and *TeTX*-treated chicks: The left-right phase.

**Figure supplement 1—source data 1.** Analysis of left-right phase.

overextension, suggesting a postcollapse compensation in the extensor drive. The rest of the collapses (12.63% ± 7.56%) were preceded by overextension.

## Wide-base stepping

A wide-base stance is typical of an unbalanced ataxic gait. The stride width was measured between the two feet during the double stance phase of stepping. The mean stride in *TeTX*-manipulated chicks 1, 2, 4, and 5 was significantly wider than that in the control chicks, while the width in *TeTX3* was similar to that in the controls (*Table 2*).

## Variable limb movements

In stable gait, limb trajectories are consistent from stride to stride. Therefore, we compared the trajectories of knee height and angle of the TMP joint during the swing phase of stepping between control and TeTX-manipulated chicks. Plots of the knee height and TMP angle trajectories during the normalized swing in all the analyzed steps of each chick are shown superimposed in *Figure 6D and E*, respectively. These data demonstrate that the range of changes in *TeTX*-manipulated chicks was higher than that in control chicks.

Further analyses revealed that, overall, the control group showed lower knee height and TMP angle ranges than the TeTX-treated group, even though there were differences within groups (*Figure 7—figure supplement 1*). The average knee height range of the combined control chicks (1.981 ± 0.33) was significantly lower than the range of the combined TeTX-treated chicks (3.109 ± 0.74) (*Figure 7B*). A similar comparison of the combined ranges of angular excursions of the TMP joint during the normalized swing revealed that the average angle of the control group (49.34 ± 16.03) was significantly lower than the average of the TeTX-treated chicks (77 ± 22.15; *Figure 7C*).

Since the increased range of changes could be due to the effects of the substantial increase in body collapses during stepping (*Figure 7A*, see also *Figure 6*), we excluded steps featuring whole-body collapses and reanalyzed the data. The data summarized in *Table 3* show that the significant difference between controls and the *TeTX*-treated chicks in the range of the knee height and the TMP angle excursions was maintained. Thus, the increase in irregularity in the *TeTX*-treated chicks is not caused exclusively by the body collapses of the *TeTX*-treated chicks.

Loss of balance can also arise from slipping, which can originate from a shallow landing angle (*Clark and Higham, 2011*). The landing angle was characterized as the angle between the ground and the imaginary line connecting the knee joint (which is located near the chicken's center of mass; *Daley and Biewener, 2006*) to the TMP joint at the end of the swing phase (*Figure 7—figure supplement 2A*). Thus, we analyzed the landing angles of the manipulated and control chickens. No significant differences in landing angles were detected (*Figure 7—figure supplement 2B*). Additionally, we compared the landing angle before a collapse to landing angles not preceding a collapse. We found that the angles preceding a collapse were not smaller and even tended to be slightly larger than the landing angles that did not precede a collapse (*Figure 7—figure supplement 2C*, p=0.02). These results argue against the possibility that the increased frequency of collapses in the manipulated chickens stemmed from slipping and sliding events.

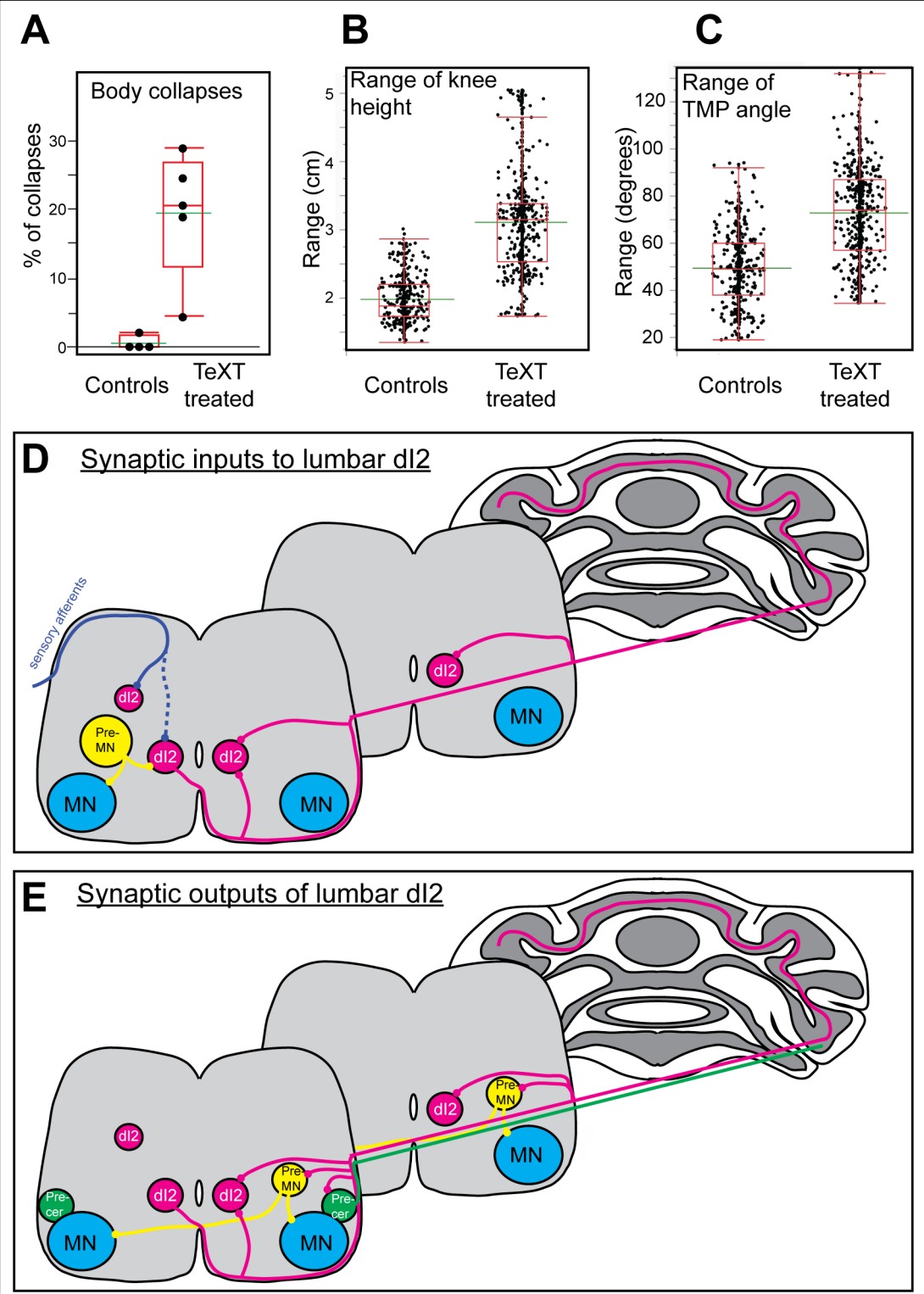

**Figure 7.** Parameters of reduced stability in bipedal stepping in *TeTX*-treated chicks. (**A**) The percentage of steps with body collapses in the controls and *TeTX*-manipulated hatchlings (n = 4 and n = 5, respectively). p-value<0.0001 (Z-test). See Table 3 for the proportions of falls at the individual chick level. (**B**) Analysis of the mean range of knee height changes during the swing phase of control and TeTX-treated chicks (n = 4 and n = 5, respectively). p-value<0.0001 using a t-test allowing different variances. See *Figure 7—figure supplement 1A* and Table 3 for individual chick data and statistical analysis details. (**C**) Analysis of the mean range of TMP angular excursions during the swing phase of control- and TeTX-treated chicks (n = 4 and n = 5, respectively). p-value<0.0001, Watson–Williams test. See *Figure 7—figure supplement 1B* and Table 3 for individual chick data and statistical

*Figure 7 continued on next page*

*Figure 7 continued*

analysis details. (**D, E**) Schematic illustrations showing the connectome of lumbar dI2 neurons. The synaptic inputs (**D**) and outputs (**E**) of dI2 neurons are illustrated. dI2 neurons (magenta) receive synaptic input from sensory afferents (solid blue line indicates massive synaptic input, and dashed blue line indicates sparse innervation), from inhibitory and excitatory premotor neurons (pre-MNs; yellow), and from the contralateral lumbar dI2 neurons. dI2 neurons innervate the contralateral lumbar and brachial pre-MNs (both commissural and ipsilaterally projecting pre-MNs are innervated by dI2 cells), the lumbar and brachial contralateral dI2 cells, lumbar precerebellar neurons (green), and the cerebellar granule cells. See *Figure 7—source data 1*.

The online version of this article includes the following source data and figure supplement(s) for figure 7:

**Source data 1.** Analysis of collapses.

**Figure supplement 1.** Locomotion characteristics of control and *TeTX*-treated chicks: The range of knee height and TMP angles.

**Figure supplement 1—source data 1.** Statistical analysis of knee height trajectories and TMP angles of each chick.

**Figure supplement 2.** Locomotion characteristics of control and *TeTX*-treated chicks.

**Figure supplement 2—source data 1.** Analysis of the landing angles in all steps.

**Figure supplement 2—source data 2.** Analysis of the landing angles prior to a collapse.

Overall, the kinematic parameters of the dI2-TeTX-treated chicks demonstrate a reduction in stability during locomotion, indicating a possible role of dI2 in the stabilization of bipedal stepping.

## Discussion

The VSCT is thought to provide peripheral and intrinsic spinal information to the cerebellum to shape and update the output of spinal networks that execute motor behavior. The lack of genetic access to VSCT neurons hampers efforts to elucidate their role in locomotion. Using a genetic toolbox to dissect the circuitry and manipulate neuronal activity in the chick spinal cord, we studied spinal INs with VSCT characteristics. The main finding in our study is that dI2 neurons in the chick lumbar spinal cord are commissural neurons that innervate pre-MNs at the contralateral lumbar and brachial spinal levels and granule neurons in the ipsilateral cerebellum. Hence, a subpopulation of dI2 neurons form part of the avian VSCT. Targeted silencing of dI2 neurons leads to impaired stepping in P8 hatchlings. We described the spatial distribution of subpopulations of dI2 neurons, deciphered their connectomes, mapped the trajectory of their projections to the cerebellum, and suggested possible mechanisms for the gait perturbation resulting from their genetic silencing, as discussed below.

### The connectome of dI2 neurons

Using the intersection between genetic drivers and spatially restricted delivery of reporters to define lumbar and brachial neurons, we identified several targets of dI2 lumbar neurons. Lumbar dI2 neurons innervate contralateral lumbar dI2 neurons as well as commissural and noncommissural lumbar pre-MNs. This connectivity may influence the bilateral spinal output circuitry at the lumbar cord (e.g., *Bras et al., 1988*; *Jankowska and Hammar, 2013*). Moreover, the ascending axons of lumbar dI2 neurons give off gray matter collaterals innervating contralateral dI2 neurons and commissural and noncommissural pre-MNs throughout the brachial spinal cord (*Figure 7D and E*). Therefore, lumbar dI2 neurons may also contribute to the inter-enlargement coupling described between the segments of the spinal cord that move the legs and the wings (e.g., *Valenzuela et al., 1990*; *Ruder et al., 2016* for forelimb to hindlimb coupling connectivity in mice).

We demonstrated that lumbar dI2 cells receive sensory innervation, premotor inhibitory and excitatory innervation, and innervation from contralateral lumbar dI2 cells (*Figure 7D and E*). Thus, lumbar dI2 neurons can provide the cerebellum and the contralateral pre-MNs with proprioceptive information, copies of motor commands delivered from the ipsilateral pre-MNs, and integrated information from contralateral dI2 neurons (*Figure 7D and E*).

Our wiring-decoding studies are based on the availability of enhancer elements that direct expression in specific spinal INs. The lack of enhancer elements for known pre-MNs, such as V2, precluded their analysis. Future experiments using identified regulatory elements that direct expression in two other pre-MN populations, the dI3 and V0 neurons (*Avraham et al., 2010b*; *Gard et al., 2017*), will reveal the extent of premotor information relayed by dI2 neurons.

**Table 2.** Maximum stride width in control and tetanus toxin (TeTX)-manipulated chicks.
Stride width was measured as described in Materials and methods. Significance was tested using one-way ANOVA followed by Dunnett's test.

| Chick | Maximum stride width (cm) | # of steps |
|---|---|---|
| TeTX1 | 5.11 ± 1.89 | 97 |
| TeTX2 | 5.32 ± 1.38 | 36 |
| TeTX3 | 4.5 ± 1.01 | 27 |
| TeTX4 | 4.9 ± 1.16 | 49 |
| TeTX5 | 5.82 ± 1.71 | 110 |
| Control 8 | 4.15 ± 1.07 | 137 |
| Control 9 | 4.32 ± 1.32 | 115 |

## dI2 subpopulations

Our study reveals two anatomically distinct subpopulations of dI2 neurons: precerebellar projection dI2 cells, which also innervate spinal targets along the entire extent of the spinal cord, and propriospinal dI2 cells, which innervate targets within the lumbar level. The laminar and medial/lateral positions of the two dI2 populations are similar, and we did not find subtype-specific expression of the known dI2 TFs. Contrary to our findings, a recent study (*Osseward et al., 2021*) has shown that tract and propriospinal INs in the mouse spinal cord differ in the localization of their somata along the mediolateral axis and their transcription of TFs. Tract neurons reside in the lateral spinal cord and express group-N TFs, while propriospinal neurons settle at the medial spinal cord and express group-Z TFs. Several reasons may explain the discrepancy between our results and those of Osseward et al. The analysis in mice (*Osseward et al., 2021*) was applied to numerous cardinal populations of neurons but excluded dI2 neurons. Hence, dI2 may represent an exception. In addition, our study revealed that precerebellar dI2 neurons share two wiring patterns: tract and propriospinal neurons. Applying single-cell RNA sequencing to dI2 neurons will reveal whether short- and long-range targeting in dI2 neurons are characterized by distinct transcriptomes or by a shared N + Z transcriptome.

## Physiological role of dI2 neurons

The unclear genetic origin of physiologically equivalent lumbar VSCT neurons has prevented a better understanding of their role in hindlimb locomotion. Our wiring and neuronal-silencing studies implicated dI2 as a significant contributor to the regularity and stability of locomotion in P8 hatchlings. The kinematic analysis of TeTX-treated hatchlings revealed imbalanced locomotion with occasional collapses, increased stride variability, a wide-base gait, and variable limb movements during stepping.

The mechanisms accounting for impaired stepping following dI2 neuron silencing are still unknown. One of the possible mechanisms is that silencing dI2 neurons perturbs the delivery of peripheral and intrinsic feedback to the cerebellum, leading to unreliable updating of the motor output produced by the locomotor networks, thereby impairing bipedal stepping. Another possible mechanism is based on the similarity of the gait instabilities of TeTX-treated hatchlings to ataxic motor disorders. Mammalian VSCT neurons receive descending input from reticulospinal, rubrospinal, and vestibulospinal pathways (*Bras et al., 1988*; *Jankowska and Hammar, 2013*). Neurons from the lateral vestibular nucleus have been reported to innervate extensor motoneurons at the lumbar level, as well as INs residing at medial lamina VII (*Murray et al., 2018*), the location where dI2 neurons were found to reside in our study. Thus, the vestibulospinal tract may convey input directly to the ipsilateral motor neurons and indirectly to contralateral motor neurons through dI2 neurons that innervate contralateral pre-MNs.

The local spinal connections between dI2 neurons and the contralateral pre-MNs and contralateral dI2 neurons may serve an important component of coordinated limb movements. dI2 synapses were found on both ipsilaterally and contralaterally projecting pre-MNs, both within their segmental level and at the brachial level, which regulates the movement of the wings. Thus, dI2 neurons may affect the motor output of the contralateral and ipsilateral sides of the cord by contacting commissural pre-MNs. Specific targeting of dI2 subpopulations – the precerebellar versus the propriospinal dI2 cells – is necessary to determine the relative contribution of dI2 subpopulations to the impaired stepping phenotype. However, there is no available genetic technique for differentially targeting the two subpopulations. In addition, the fact that the precerebellar dI2 neurons also innervate the lumbar spinal cord precludes the use of retrogradely target-derived neuronal activity modifiers.

In summary, our mapping studies of dI2 neurons and their connectomes, followed by characterization of the effects of their silencing on bipedal stepping, offer new insights into the function of

**Table 3.** Collapses, knee height, and TMP angle ranges in control and tetanus toxin (TeTX)-manipulated chicks.

Analysis of the range between the highest and lowest knee heights and the largest and smallest angles of the TMP joint in all steps before and after subtraction of collapsed steps. In the combined results, the difference was statistically significant (p-value<0.0001 for both metrics, using a t-test allowing different variances).

| Chick | % of steps with collapse | TMP angle: mean range (°) | | | | # of steps | | | | % of steps with collapse |
|---|---|---|---|---|---|---|---|---|---|---|
| | | All steps, mean | Combined mean | Minus collapses, mean | Minus collapses, combined mean | All steps, mean | Combined mean | Minus collapses | Minus collapses, combined mean | |
| TeTX1 | 4.4 | 3.05 ± 0.49 | | 2.87 ± 0.38 | | 82.27 ± 22 | | 79.4 ± 24.5 | | 113 |
| TeTX2 | 20.60 | 3.5 ± 0.3 | | 3.35 ± 0.47 | | 71.79 ± 25 | | 71.1 ± 25.68 | | 63 |
| TeTX3 | 18.8 | 2.57 ± 0.31 | | 2.27 ± 0.26 | | 64.17 ± 21 | | 62.79 ± 20.9 | | 69 |
| TeTX4 | 20.45 | 2.54 ± 0.55 | | 2.477 ± 0.56 | | 72.48 ± 17 | | 66.22 ± 19.1 | | 59 |
| TeTX5 | 29 | 3.86 ± 0.83 | 3.11 ± 0.74 | 3.2 ± 0.2 | 2.83 ± 0.57 | 72.86 ± 12 | 72.71 ± 20.52 | 65.85 ± 13.13 | 68.97 ± 21.47 | 93 |
| Control 1 (GFP) | 2.12 | 1.91 ± 0.22 | | 1.91 ± 0.22 | | 56.8 ± 16.4 | | 56.95 ± 16.6 | | 47 |
| Control 2 (GFP) | 0 | 1.83 ± 0.25 | | 1.83 ± 0.25 | | 41.42 ± 18.4 | | 41.42 ± 18.4 | | 19 |
| Control 3 (not electroporated) | 0 | 2.42 ± 0.23 | | 2.42 ± 0.23 | | 54.86 ± 9.65 | | 54.86 ± 9.65 | | 121 |
| Control 4 (not electroporated) | 0 | 1.75 ± 0.1 | 1.98 ± 0.33 | 1.75 ± 0.1 | 1.98 ± 0.33 | 44.12 ± 12.51 | 49.34 ± 16.03 | 44.12 ± 12.51 | 49.4 ± 16.18 | 51 |

dl2 neurons in vertebrates. We suggest that lumbar dl2 neurons not only relay sensory and intrinsic spinal network information to the cerebellum but also act as active mediators of motor functions at the lumbar segments and at the wing-controlling brachial segments of the spinal cord. Further circuit-deciphering studies of the constituents of subpopulations of dl2 cells, their targets, and their descending inputs are required to extend our understanding of the function of dl2 subpopulations in motor control.

# Materials and methods

### Key resources table

| Reagent type (species) or resource | Designation | Source or reference | Identifiers | Additional information |
|---|---|---|---|---|
| Strain, strain background (Chicken) | *Gallus gallus* | Gil-Guy Farm, Israel | NCBI Taxon: 9031 | |
| Strain, strain background (*Pseudorabies virus*) | PRV152 | *Enquist and Card, 2003* | NCBI Taxon: 10345 | |
| Strain, strain background (*Pseudorabies virus*) | PRV614 | *Enquist and Card, 2003* | NCBI Taxon: 10345 | |
| Antibody | Rabbit anti-GFP (polyclonal) | Molecular Probes, Eugene, Oregon, USA | A-11122 RRID:AB_221569 | Dilution (1:1000) |
| Antibody | Mouse anti-GFP (monoclonal) | Abcam | Ab1218 AB_298911 | Dilution (1:100) |
| Antibody | Goat anti-GFP (polyclonal) | Abcam | Ab6673 RRID:AB_305643 | Dilution (1:300) |
| Antibody | Rabbit anti-RFP (polyclonal) | Acris | AP09229PU-N RRID:AB_2035909 | Dilution (1:1000) |
| Antibody | Goat anti-ChAT (polyclonal) | Millipore, USA | AB144P RRID:AB_2079751 | Dilution (1:300) |
| Antibody | Mouse anti-synaptotagmin (monoclonal) | Hybridoma Bank, University of Iowa, Iowa City, USA | ASV30 RRID:AB_2295002 | Dilution (1:100) |
| Antibody | Mouse anti-lhx1/5 (monoclonal) | Hybridoma Bank, University of Iowa, Iowa City, USA | 4F2 RRID: AB_531784 | Dilution (1:100) |
| Antibody | Mouse anti-FoxP4 (monoclonal) | Hybridoma Bank, University of Iowa, Iowa City, USA | PCRP-FOXP4-1G7 RRID:AB_2618641 | Dilution (1:50) |
| Antibody | Rabbit anti-Pax2 (polyclonal) | Abcam | ab79389 RRID:AB_1603338 | Dilution (1:50) |
| Antibody | Chicken anti- lacZ (polyclonal) | Abcam | ab79389 RRID:AB_307210 | Dilution (1:300) |
| Antibody | Rabbit anti-calbindin (polyclonal) | Swant | D-28k RRID:AB_2314070 | Dilution (1:200) |
| Antibody | Goat anti-FoxP2 (polyclonal) | Abcam | ab1307 RRID:AB_1268914 | Dilution (1:1000) |
| Antibody | Rabbit anti-5-HT (polyclonal) | Abcam | ab140495 | Dilution (1:100) |
| Recombinant DNA reagent | Edl1::Cre | *Avraham et al., 2009* | N/A | |
| Recombinant DNA reagent | Ngn1::Cre | *Avraham et al., 2009* | N/A | |
| Recombinant DNA reagent | Ngn1::FLPo | *Hadas et al., 2014* | N/A | |

*Continued on next page*

*Continued*

| Reagent type (species) or resource | Designation | Source or reference | Identifiers | Additional information |
|---|---|---|---|---|
| Recombinant DNA reagent | Foxd3::FLPo | *Hadas et al., 2014* | N/A | |
| Recombinant DNA reagent | Foxd3::Cre | *Avraham et al., 2009* | N/A | |
| Recombinant DNA reagent | Isl1::Cre | *Avraham et al., 2010a* | N/A | |
| Recombinant DNA reagent | CAG-LSL-GFP | *Hadas et al., 2014* | N/A | |
| Recombinant DNA reagent | CAG-LSL-SV2-GFP | *Hadas et al., 2014* | N/A | |
| Recombinant DNA reagent | CAG-FSF-LSL-GFP | *Hadas et al., 2014* | N/A | |
| Recombinant DNA reagent | CAG-FSF-LSL-SV2-GFP | This paper | N/A | *Figure 1—figure supplement 1*; can be obtained from the Klar lab |
| Recombinant DNA reagent | CAG-FSF-LSL-cherry | This paper | N/A | *Figure 1—figure supplement 1*; can be obtained from the Klar lab |
| Recombinant DNA reagent | CAG-FSF-LSL-SV2-cherry | This paper | N/A | *Figure 1—figure supplement 1*; can be obtained from the Klar lab |
| Recombinant DNA reagent | CAG-FSF-LSL-AP | This paper | N/A | *Figure 1—figure supplement 1*; can be obtained from the Klar lab |
| Recombinant DNA reagent | CAG-LSL-TeXT | This paper | N/A | *Figure 1—figure supplement 1*; can be obtained from the Klar lab |
| Recombinant DNA reagent | CAG-LSL-F_SV2-cherry_F-GFP | This paper | N/A | *Figure 1—figure supplement 1*; can be obtained from the Klar lab |
| Recombinant DNA reagent | pGEMTEZ-TeTxLC | Addgene | #32640 | |
| Sequence-based reagent | Foxd3-F | This paper | PCR primers | TCATCACCATGGCCATCCTG |
| Sequence-based reagent | Foxd3-R | This paper | PCR primers | GCTGGGCTCGGATTTCACGAT |
| Sequence-based reagent | vGlut2-F | This paper | PCR primers | GGAAGATGGGAAGCCCATGG |
| Sequence-based reagent | vGlut2-R | This paper | PCR primers | GAAGTCGGCAATTTGTCCCC |
| Sequence-based reagent | VIAAT-F | This paper | PCR primers | CTGAACGTCACCAACGCCATCC |
| Sequence-based reagent | VIAAT-R | This paper | PCR primers | GGGTAGGAGAGCAAGGCTTTG |
| Commercial assay or kit | NucleoBond Xtra Midi | Macherey-Nagel | Cat # 740410.50 | |
| Chemical compound, drug | CTB conjugated to Alexa Fluor 647 | Thermo Fisher | C34778 | 0.3 M |
| Software, algorithm | JMP | JMP | https://www.jmp.com/en_gb/home.html | |
| Software, algorithm | Adobe Photoshop | Adobe | https://www.adobe.com/il_en/ | |
| Software, algorithm | ImageJ | ImageJ | https://imagej.nih.gov/ij/ | |
| Software, algorithm | IMARIS | Oxford Instruments | https://imaris.oxinst.com/ | |
| Software, algorithm | MacVector | MacVector | https://macvector.com/index.html | |

*Continued on next page*

*Continued*

| Reagent type (species) or resource | Designation | Source or reference | Identifiers | Additional information |
|---|---|---|---|---|
| Other (electroporator) | BTX Electroporator | BTX Harvard Apparatus | Cat#45-0662 | |
| Other (confocal microscope) | FV1000; Olympus | Olympus | https://www.olympus-global.com/ | |
| Other (microscope) | Eclipse Ni | Nikon | https://www.nikon.com/ | |
| Other (light-sheet microscope) | LaVision Ultramicroscope II light-sheet microscope | LaVision BioTec | https://www.lavisionbiotec.com/ | |

## Animals

Fertilized white leghorn chicken eggs (Gil-Guy Farm, Israel) were incubated under standard conditions at 38 °C. All experiments involving animals followed the designated policies of the Experiments in Animals Ethics Committee and were performed with its approval.

## 3D reconstruction and density plot analysis

The programs for both 3D reconstruction and the density plot analysis were written in MATLAB. The density plots were generated based on cross-sectional images converted to a standard form. The background was subtracted, and the cells were filtered automatically based on their soma size or using a manual approach. Subsequently, two-dimensional kernel density estimation was obtained using the MATLAB function '*kde2d*.' Finally, unless indicated otherwise, a contour plot was drawn for density values between 20% and 80% of the estimated density range, with six contour lines.

## In ovo electroporation

A DNA solution of 5 mg/mL was injected into the lumen of the neural tube at HH stage 17–18 (E2.75–E3). Electroporation was performed using 3 × 50 ms pulses at 25–30 V, applied across the embryo using a 0.5 mm tungsten wire and a BTX electroporator (ECM 830). Following electroporation, 150–300 µL of antibiotic solution containing 100 unit/mL penicillin in Hanks' balanced salt solution (Biological Industry, Beit-Haemek) was added on top of the embryos. Embryos were incubated for 3–19 days prior to further treatment or analysis.

## Immunohistochemistry and in situ hybridization

Embryos were fixed overnight at 4 °C in 4 % paraformaldehyde/0.1 M phosphate buffer, washed twice with phosphate buffered saline (PBS), incubated in 30 % sucrose/PBS for 24 hr, and embedded in optimal cutting temperature (OCT) compound (Scigen, Grandad, USA). Sections with a thickness of 20 µm were cut on a cryostat. These sections were collected on Superfrost Plus slides and kept at −20 °C. For 100 µm sections, spinal cords were isolated from the fixed embryos and subsequently embedded in warm 5 % agar (in PBS), and 100 µm sections (E12–E17) were cut with a vibratome. Sections were collected in wells (free-floating technique) and processed for immunolabeling.

The following primary antibodies were used: rabbit polyclonal GFP antibody 1:1000 (Molecular Probes, Eugene, OR, USA), mouse anti-GFP 1:100, goat anti-GFP 1:300 (Abcam), rabbit anti-RFP 1:1000 (Acris), goat anti-ChAT antibody 1:300 (Cemicon, Temecula, CA, USA), mouse anti-synaptotagmin antibody 1:100 (ASV30), mouse anti-Lhx1/5 1:100 (4F2), mouse anti-FoxP4 1:50 (hybridoma bank, University of Iowa, Iowa City, USA), mouse anti-Brn3a 1:50 (Mercury), rabbit anti-Pax2 antibody 1:50 (Abcam), chicken anti-lacZ antibody 1:300 (Abcam), rabbit anti-Calbindin 1:200 (Swant), rabbit anti-VGLUT2 antibody (Synaptic Systems, Göttingen, Germany), goat anti-FoxP2:1000 (Abcam), anti-FoxP1:100 (ABR Synaptic), and rabbit anti-5-HT (Abcam). The following secondary antibodies were used: Alexa Fluor 488/647-conjugated AffiniPure donkey anti-mouse, anti-rabbit, and anti-goat (Jackson) and Rhodamine Red-X-conjugated donkey anti-mouse and anti-rabbit (Jackson). Images were taken under a microscope (Eclipse Ni; Nikon) with a digital camera (Zyla sCMOS; Andor) or captured using the integrated camera of a confocal microscope (FV1000; Olympus).

In situ hybridization was performed as previously described (*Avraham et al., 2010a*). The following probes were employed: Foxd3, vGlut2, and VIAAT probes were amplified from the cDNA of E6 chick

embryos using the following primers. Foxd3: forward TCATCACCATGGCCATCCTG, reverse GCTG GGCTCGGATTTCACGAT. vGlut2: forward GGAAGATGGGAAGCCCATGG, and reverse GAAGTCG-GCAATTTGTCCCC. VIAAT: forward CTGAACGTCACCAACGCCATCC, reverse GGGTAGGAGAGC AAGGCTTTG. The T7 RNA polymerase cis-binding sequence was added to the reverse primers.

## Laminar division

The standard forms of the spinal cord (for the crural, sciatic, and brachial plexus levels) were computationally divided into polygons for the different laminae (*Martin, 1979*). The number of neurons or synapses inside each lamina border was scored using their coordinates.

Light-sheet microscopy dI2::mCherry was electroporated into the embryos at HH stage 17–18. Embryos were removed at E13, and the spinal cord and cerebellum were isolated. The tissue was cleared using the iDISCO technique as described (*Renier et al., 2014*). The mCherry-expressing neurons were stained by application of an anti-RFP antibody followed by Rhodamine Red-X-conjugated donkey secondary antibody. Each staining step included 3 days of incubation with the antibody and subsequent washing for 2 days. Then, the cleared tissue was divided into three segments: a lumbar spinal segment, a brachial spinal segment, and a segment including the brainstem and cerebellum. Each sample was placed in a quartz imaging chamber (LaVision BioTec) and scanned by a LaVision Ultramicroscope II light-sheet microscope operated by ImspectorPro software (LaVision BioTec). An Andor Neo sCMOS camera was used for 16-bit image acquisition. The imaging was performed at 2× magnification with a 0.5–1 µm step size and a green excitation filter (peak – 525 nm/width – 50 nm). Then, for 3D reconstruction and analysis of the samples, the resulting image z-stacks were converted to IMS format using Imaris File Converter (version 9.5). Because of the sample size, several z-stacks were required for full acquisition of each sample; they were stitched together into one z-stack by Imaris Stitcher. Then, the files were uploaded to Imaris (9.6 version) for advanced visualization and analysis. dI2 axons were tracked using the filament tracer feature in semiautomatic mode. After tracking, Imaris was used to generate videos and snapshots describing different features of the analyzed samples. Finally, text, arrows, and other symbols were added using Adobe AfterEffects software.

## Synaptic marker validation

Validation of SV2-GFP reporter specificity was performed by using Imaris software. High-resolution confocal images of spinal cord sections, with clear SV2-GFP reporter expression and synaptotagmin (syn) immunolabeling, were used to quantify the degree of overlap of GFP$^+$ terminals and syn$^+$-labeled boutons. Both signals were three-dimensionally reconstructed, and we used the automatic quantification abilities of Imaris, further validated by additional manual counting, to quantify the number of GFP$^+$ presynaptic terminals containing at least one syn$^+$ bouton. In addition, the volume of GFP$^+$ presynaptic terminals was documented to explore a possible dependence between terminal volume and syn$^+$ bouton containment.

## AP staining

The treated embryos were fixed with 4 % paraformaldehyde–PBS for 24 hr at 4 °C and washed twice with PBS for 30 min at 4 °C. The fixed embryos were incubated at 65 °C in PBS for 8–16 hr to inactivate endogenous AP activity. The treated embryos were washed with 100 mM Tris–Cl (pH 9.5) containing 100 mM NaCl and 50 mM $MgCl_2$, and the residual placental alkaline phosphatase activity was visualized by incubating the embryos with NBT/BCIP (Roche) in the same buffer at 4 °C for 24 hr. After extensively washing the embryos with PBS–5 mM EDTA, the spinal cord was imaged.

## PRV infection and CTB retrograde labeling

From the attenuated PRV Bartha strain, we used two isogenic recombinants that express enhanced GFP (PRV152) and monomeric red fluorescent protein (PRV614). The viruses were harvested from Vero cell cultures at titers of $4 \times 10^8$, $7 \times 10^8$, and $1 \times 10^9$ plaque-forming units (PFU/mL). Viral stocks were stored at −80 °C. Injections of 3 µL of PRV152 or PRV614 were made into the thigh, pectoralis, or distal wing musculature of E13 or E14 chick embryos using a Hamilton syringe (Hamilton; Reno, NV, USA) equipped with a 33-gauge needle. The embryos were incubated for 36–40 hr and sacrificed for analysis. For spinocerebellar projecting neuron labeling, we used a replication-defective HSV (TK⁻) that contains a lacZ reporter. The virus was injected into the cerebellum of E12–15 embryos in ovo,

and the embryos were incubated for another 40–48 hr. Alternatively, CTB conjugated to Alexa Fluor 647 (Thermo Fisher) was injected into the cerebellum of E12–15 embryos together with the virus for visualization of both precerebellar neurons and the upstream neurons.

### Force test

Muscle strength was evaluated using the measurement of the slope at which the chicks fell from a mesh surface as it was gradually tilted up from the horizontal. This test was repeated for each chick at least three times, and the average falling angle was calculated.

### Analysis of left-right phase

Stride duration was measured as the time from right toe-off/foot-off to the next right toe-off (as a complete stride cycle for the right leg), and the 'half-cycle' duration was measured as the time from right toe-off to the time of left toe-off. The following formula was used to calculate the phase: ((Left-ToeOff_1 - RightToeOff_1)/(RightToeOff_2 – RightToeOff_1))*360.

### Behavioral tests and analysis

The embryos were bilaterally electroporated and then allowed to develop and hatch in a properly humidified and heated incubator. Afterwards, within 32 hr after hatching, the hatchling chicks were imprinted on the trainer. The P8 chicks were filmed in slow motion (240 fps) while freely walking (side and top views). The following parameters were scored: (1) weight; (2) foot grip strength; and (3) kinematic parameters during overground locomotion: (a) swing velocity, (b) swing and stance duration, (c) phase of footfalls, (d) heights of the knee and TMP joints, (e) angles of the TMP and ankle joints, (f) stride width (distance between feet during the double stance phase), and (g) landing angle.

Using semiautomated MATLAB-based tracking software (*Hedrick, 2008*), several points of interest were encoded. The leg joints as well as the eye and the tail were tracked. The position of these reference points was used for computational analysis using in-house MATLAB code for calculating different basic locomotion parameters (e.g., stick diagrams, velocity, joint trajectory, angles, range, and elevation), step patterns, and degrees of similarity. The landing angle was calculated as the angle between the imaginary line connecting the knee and the TMP joints and the ground, at the end of the swing phase. Dunnett's test (*Dunnett, 1955*) was used to perform multiple comparisons of group means following one-way ANOVA. Circular statistics were used for analyses of angular data utilizing Oriana (KCS, version 4).

## Acknowledgements

The authors thank Haya Falk for PRV purification; Alona Katzir, Cole Bendor, Mevaseret Avital, Sapir Shevah, Eitan Yisraeli, Ruth Segal, Fedaa Bazan, and Eden Kimchi for technical assistance; Nadav Yayon for assistance with the light sheet microscopy; and Michael O'Donovan for comments on the manuscript. This work was supported by grants to AK from the Israel Science Foundation (grant no. 1400/16), the US–Israel Binational Science Foundation (grant no. 2017/172), and the Avraham and Ida Baruch endowment fund.

## Additional information

### Funding

| Funder | Grant reference number | Author |
| --- | --- | --- |
| Israel Science Foundation | 1400/16 | Avihu Klar |
| United States - Israel Binational Science Foundation | 2017/172 | Avihu Klar |
| The Avraham and Ida Baruch Endowment Fund | | Avihu Klar |

| Funder | Grant reference number | Author |
| --- | --- | --- |

The funders had no role in study design, data collection and interpretation, or the decision to submit the work for publication.

## Author contributions

Baruch Haimson, Conceptualization, Data curation, Formal analysis, Investigation, Methodology, Software, Writing - original draft, Writing - review and editing; Yoav Hadas, Data curation, Formal analysis, Investigation, Methodology; Nimrod Bernat, Methodology; Artur Kania, Conceptualization, Writing - review and editing; Monica A Daley, Conceptualization, Data curation, Writing - original draft; Yuval Cinnamon, Conceptualization, Data curation, Methodology; Aharon Lev-Tov, Conceptualization, Formal analysis, Investigation, Methodology, Supervision, Validation, Writing - original draft, Writing - review and editing; Avihu Klar, Conceptualization, Data curation, Formal analysis, Funding acquisition, Investigation, Methodology, Resources, Supervision, Validation, Writing - original draft, Writing - review and editing

## Author ORCIDs

Baruch Haimson [ID] http://orcid.org/0000-0002-0163-6196
Artur Kania [ID] http://orcid.org/0000-0002-5209-2520
Aharon Lev-Tov [ID] http://orcid.org/0000-0002-3906-0057
Avihu Klar [ID] http://orcid.org/0000-0002-9248-2179

## Ethics

All experiments involved with animals were conducted in accordance with the designated Experiments in Animals Ethic Committee policies and under its approval.

## Decision letter and Author response

Decision letter https://doi.org/10.7554/eLife.62001.sa1
Author response https://doi.org/10.7554/eLife.62001.sa2

# Additional files

## Supplementary files

• Supplementary file 1. Weight, force, and number of electroporated cells. See statistical tests in Supplementary Statistical analysis tables.

• Transparent reporting form

## Data availability

All data generated or analysed during this study are included in the manuscript and the supporting files.

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
