## [Decision Letter]

**Acceptance summary:**

The study by Haimson and colleagues investigates, in the chick model, the role of different sub-classes of developmentally-characterized dI2 interneurons in coordinating activity across different spinal regions, regulating stepping and reporting back activity to the brain. The work brings new insights in the so far under-investigated spinal dorsally-born neuronal types and about long-range projecting neurons that link the spinal cord with higher integrative centers.

**Decision letter after peer review:**

Thank you for submitting your article "Spinal dI2 interneurons regulate the stability of bipedal stepping" for consideration by *eLife*. Your article has been reviewed by 3 peer reviewers, and the evaluation has been overseen by a Reviewing Editor and Ronald Calabrese as the Senior Editor. The following individuals involved in review of your submission have agreed to reveal their identity: David SK Magnuson (Reviewer #1); Julien Bouvier (Reviewer #2).

The reviewers have discussed the reviews with one another and the Reviewing Editor has drafted this decision to help you prepare a revised submission.

We would like to draw your attention to changes in our revision policy that we have made in response to COVID-19 (https://elifesciences.org/articles/57162). Specifically, when editors judge that a submitted work as a whole belongs in eLife but that some conclusions require a modest amount of additional new data, as they do with your paper, we are asking that the manuscript be revised to either limit claims to those supported by data in hand, or to explicitly state that the relevant conclusions require additional supporting data.

Our expectation is that the authors will eventually carry out the additional experiments and report on how they affect the relevant conclusions either in a preprint on bioRxiv or medRxiv, or if appropriate, as a Research Advance in eLife, either of which would be linked to the original paper.

Summary:

The three reviewers and I have carefully read your manuscript and we all have found a great interest in your results. However several concerns will have to be addressed in order to consider the paper further. In a first instance the paper requires careful revisions in the writing/editing, quantifications and clarity of the figures. Also the discussion on the main findings and their limitations should be revised in order to take into account reviewers' comments.

More specifically, the authors are asked to be more accurate/careful in describing:

1) The motor phenotype,

2) The identity of which sub-population of neurons is considered to be responsible for the locomotor behavior (di2 neurons vs vSCT neurons),

3) The anatomical data of the neurons of interest, possibly including their neurotransmitter phenotype,

4) The quantification and characterization of the synaptic contacts and their potential functional role. To address some of these issues additional experiments might be required, as more-specific silencing studies for example, but this might not be always necessary if the text is changed accordingly and if potential alternative explanations are given whenever possible.

Overall, we agreed that the paper has some important contributions to make if appropriately modified. Its suitability for publication will be re-evaluated after revisions.*Reviewer #1:*

This is a well-put-together manuscript describing carefully performed circuitry dissection and functional analysis of dl2 neurons in the chick. A genetic toolbox is used taking advantage of the electroporation technique applied to the embryos. The findings include a fairly convincing connectome for dl2 neurons and a functional phenotype that is, unfortunately, rather unsatisfying. The investigators conclude that dl2 interneurons regulate "stability" of bipedal stepping in the chick, which is fine, but the analysis misses an opportunity to more fully explore what the instability involves and thus to perhaps shed more light on the likely roles of this neuron population. The concerns/issues 3 and 4 below focus on this issue and the need for additional careful analysis of the behavior that will allow the phenotype to be more precisely described or ascribed to some aspect of stepping that might guide future studies in other models. For example, can the link between partial collapse and over-extensions be made more solid and thus argue that reduced extensor gain might be what results in the instability? What other analysis could be performed using the existing data/video to better describe the behavioral phenotype?

Concerns/issues:

1. The connectome part of the work appears solid and supports the concept that a sub population of the population are likely VSCT neurons, that the non VSCT neurons receive the bulk of the afferent input and that these neurons project to contralateral dl2 neurons (some which may be VSCT) and other premotor neurons. Anatomically, the only concern is that no distinctions were made between the lumbar and brachial populations, and if differences in these populations exist, it would be important and interesting to describe them.

2. Figure 2 Characterization of dl2/VSCT neurons as being primarily large dl2 neurons is quite convincing, and the observation that the dl2 neurons account for 10% of the VSCT axons is also of interest and quite compelling. A question arises, however, about the source, rostrocaudally, of the VSCT neurons and tract. Is the 10% for the total or for a specific level or levels? Can more be said/quantified about differences in these populations at different spinal levels?

3. Whole-body collapses and subsequent over-extensions are important and speak to changes in reflex arc and motor output. The statement "usually followed by" over-extension should be followed-up. Can this be further quantified? Are the two events linked or distinct, and did over-extensions happen in the absence of collapses?

4. These issues mesh with the lower knee height and angle of the TMP joint, even when collapses are excluded. It appears as though the control system to maintain muscle shortening (force output of extensors) is altered. I agree that stability is compromised, but could we go further to state that the compromise is due to extensor gain control?*Reviewer #2:*

This work addresses the possibility that developmentally-characterized di2 neurons contribute to the ventral spinocerebellar tract and regulate stepping in the chick. The work is sound considering that most information we have on spinal subtypes are for ventrally-born and local circuit interneurons (i.e. motor related), but less is known about the dorsally-born types and about long-range projecting neurons that link the spinal cord with higher integrative centers. Here, using a combination of cell-type specific manipulations, circuit tracing tools and kinematic analysis of gaits in the chick authors propose that spinal di2 interneurons contain multiple subgroups including a population that sends projection to the cerebellum. Silencing di2 neurons overal leads to impaired stepping.

Overall, the strategy is sound and there is potential novelty that make this paper a suitable candidate for *eLife* provided the weaknesses in the scientific demonstration listed below can be first addressed, experimentally and/or by additional analysis. Equally importantly also, the work suffers from a SEVERE lack of clarity (writing, figures, results). Both must be addressed for this paper to be considered further.

I start with the scientific weaknesses:

1. Synaptic connections rely mostly on the anatomical overlap between di2 cells and the synaptic field of their putative pre-synaptic partners. While this is indeed suggestive, it is not enough to ascertain actual synaptic connections, and even less so in a comparative manner between the different groups. Furthermore, some tracers (e.g PRVmCherry) do not seem to be under a synapse-specific promoter, so labelled elements might just as well be passing fibers. Clearer evidence of actual connections should be provided, functionally if possible or at the very least by showing clearer putative boutons onto neuronal somata/dendrites, quantifying them and quantifying differences between input cell types. Current figures (2F / 3B', C', D' / 4C, D', E', F') are not sufficiently convincing since we see only one cell and can barely detect boutons visually on some of them (not to mention that pseudo-colors keep changing, see other comment below). In addition, please consider using the term "putative" or "presumed" synapses, contacts and connections throughout the study.

2. The loss of function and gait analysis is stronger and convincingly presented. However, unless I missed it, the strategy silences all di2 neurons but cannot discriminate the contributions of the pre-cerebellar ones. This poses problems for the interpretation of the data. Since this paper is about either subpopulations of di2, or the vSCT (see other comment about general scope of the work), it would be more robust if more specific silencing was included. It is currently assumed that one likely mechanism for the disturbed gait owes to the function of di2 as precerebellar neurons (line 385, 389) but the phenotype could also, or even entirely, be due to their proprio-spinal connectivity. This is a major caveat.

On top of this, writing and data presentation MUST be substantially improved on multiple aspects:

3. Please have the manuscript deeply proofread. In addition to numerous English mistakes (missing "the", "or", plural and singulars, lots of unnecessary comas, etc…) examples of confused writing include (non-exhaustive list):

(a) Line 128: what does this phrase mean ("TF expression is redundant"…).

(b) Line 159: I don't understand here, the Di2 ascend to the cerebellum, cross the midline to the targeted di2?? To which Di2 do authors refer to here, it sounds like they are in the cerebellum, or that the ascending Di2 redescend to the spinal cord…

(c) The term targeted is in fact used alternatively and confusingly to refer to either "manipulated" cells, "synaptically-targeted" cells, there is also "targeted overground locomotion",….

(d) Stage HH18 is sometimes referred to as E3. Please be consistent throughout.

(e) When describing inputs onto di2, add "neurons" (i.e. "onto di2 neurons").

4. I would appreciate more background on di2 neurons in the introduction and why these have been investigated. Currently, most of this is given in the first paragraph of the results (lines 91-100 and also line 103). Also, it is stated first that "the role of di2 neurons is elusive due to the lack of genetic targeting means" (line 59). This contradicts the later statement that "the progenitor pdi2 expresses [various transcription factors]", and that the "post mitotic di2 are defined by…" (line 103). Please clarify what is known and not known about di2 already in the introduction.

5. Related to the above, it is not sufficiently clear what is investigated here. The genetic identity of ventral spinocerebellar neurons? Or the diversity of di2 neurons? In the way the introduction is written, it gives the impression that it is the former, but then functional investigations are not specific enough (since they are targeted to the overall di2 population, see dedicated comment later). Authors should revise to make clearer what is the scope of the work.

6. Histology Figures should be made more convincing, self-explanatory, and to a higher standard.

(a) Anatomical landmarks MUST be paced on ALL figures, e.g: the midline and minimal nuclei of the cerebellum, the deep cerebellar nuclei should be indicated in Figure S4,… Also, please give the orientation axis on ALL figures (especially the ones illustrating large territories, like 2B, 4A).

(b) Add the CTB or HSV tracer on Figure 2A and check coherence: I believe for instance that HSP is wrongly stated instead of HSV in Figure 2D and PRV is wrongly stated instead of CTB in Figure 2F (and there might be other confusions throughout).

(c) It is extremely confusing that histology pseudo-colors are sometimes changed from one related figure to the other, for unclear reasons (e.g. 2B, 2B', 2C, also 2C and S4A…). Consistency will help the reader go through all panels and figures comparatively.

(d) Figures must be addressed in proper order. This also applies to supplemental figures. Otherwise, it gives the impression we have missed something.

(e) What is the rationale for plotting the overlap in area versus volume (Figure 2H, I)? If overlap with area shows a higher percentage than with volume, does it mean that the overlap is only limited to a given A/P plane? I'm really confused about this representation and its meaning.

7. Authors should avoid relying on subjective formulations like "that reside at the lateral dorsal aspect of lamina VII". Instead, they MUST demonstrate the positioning of Di2 neurons into the different spinal laminae with some form of quantitative measurements. This is currently just an "impression" that large, precerebellar Di2 are more ventral, in lamina VII and possibly VIII but without the representation of lamina borders on figures, this information cannot be appreciated by the reader. It is all essential that these borders are depicted in Figures and neurons be quantitatively allocated to each laminae. In addition/alternatively, authors should report the average D/V position of the different subtypes and test for significant differences to make the case of different spatially-confined populations stronger.

8. FoxD3 expression on Supplemental Figure 2B is not convincing. It is also not reported in the statistics of Figure 1E. Do we have to assume that all di2 investigated here are FoxD3-positive? If so, one would need a better illustration and quantifications should be given. Otherwise, I would suggest to simply rely on literature and removing that Figure S1B which is not helping. On other panels of that supplemental Figure 2, please add arrow/arrowheads on all neurons that are or not co-labelled so we can appreciate co-labelling.

9. The demonstration that di2 are excitatory is essential. It is the title of a paragraph (line 102), thus I think that the corresponding data with the neurotransmitters (Vglut2, GAD) would deserve to be in the main Figures. Also, the chosen illustration only shows ONE double-labelled cell with Vglut2. Authors should be able to show a field of view that more convincingly conveys the message with more cells.*Reviewer #3:*

This study by Haimson et al., aims at examining the diversity of dI2 interneurons and their role in coordinating activity across difference region of the spinal cord and in reporting back activity to the brain. The results show that dI2 interneurons comprise different sub-classes based on their axonal projections, soma diameter and transmitter identity. They also show that some dI2 interneurons project rostrally from the lumbar spinal cord and make putative synaptic contacts with other dI2 interneurons in the brachial spinal cord on their way to the cerebellum. Finally, it is shown that some dI2 interneurons receive putative inputs from DRG neurons and may serve to transmit movement-related feedback. An indiscriminate silencing of dI2 interneurons results in instability of locomotion. Overall, this study reports some interesting observations by showing the heterogeneity of dI2 interneurons and their potential function. I have the following concerns:

1) 12% express Pax2 and are considered inhibitory. However, Gad is expressed in only 25% of dI2 interneurons while vGlut is expressed in 88%. These proportions suggest that there are dI2 neurons that co-express vGlut and Gad. Is this the case? Are there additional inhibitory dI2 neurons in addition to those expressing Pax2 which could explain the fact that Gad labels 25% of dI2 neurons. These points need some clarifications and discussion.

2) Of all dI2 interneurons, 91% are small diameter and 9% are large diameter neurons – large diameter neurons are mostly apparent in the lumbar spinal cord. The small and large diameter dI2 neurons cannot be differentiated by their expression of TFs, but can be distinguished by their transmitter identity? Is the proportion of small and large diameter neurons the same along the spinal cord?

3) Do all dI2 neurons receive putative synaptic contacts from DRG neurons? Unless I have missed it, it would be helpful to provide quantification of the number of small vs large diameter dI2 neurons with regard to the different putative synaptic contacts they receive from DRG neurons, dI2 and V1 interneurons.

4) Lines 218-220: It is stated that DRG putative contacts are mainly targeting dorsal dI2 neurons while ventral ones receive virtually no contacts. Since large diameter VSCT dI2 neurons are located ventrally, they do not seem to receive direct sensory information. However, the authors conclude that VSCT dI2 neurons receive sensory input (lines 227-228) and also in the Discussion. There seem to be a mismatch between the results and the conclusion drawn by the authors (lines 374-377). Unless I am missing something here, this is not consisting with the conclusions of this study. Please clarify.

5) The silencing experiments are interesting, however it is unclear which sub-class of dI2 neurons and at what level (lumbar vs brachial spinal cord or cerebellum) the observed behavioral perturbations take place. It is possible to selectively silence excitatory vs inhibitory or only VSCT neurons to provide some link between dI2 sub-classes and behavioral perturbations.

---

## [Author Response]

Summary:The three reviewers and I have carefully read your manuscript and we all have found a great interest in your results. However several concerns will have to be addressed in order to consider the paper further. In a first instance the paper requires careful revisions in the writing/editing, quantifications and clarity of the figures. Also the discussion on the main findings and their limitations should be revised in order to take into account reviewers' comments.More specifically, the authors are asked to be more accurate/careful in describing1) The motor phenotype,

In the original manuscript we described that there was a higher variability in the knee joint trajectory and TMP angle during the swing phase, a significantly higher number of collapses, and a wider base stepping in the manipulated compared to the control chickens. The swing velocity and the inter-limb phase relation during stepping in the manipulated chickens were similar to those of the un-manipulated controls. In the revised version of the manuscript, we analyzed two additional aspects of the motor phenotypes: (1) we examined whether slipping over the walking surface contributed to the observed instability (e.g. Clark and Higham, 2011, Daley and Biewener, 2006). (2) according to the request of Reviewer 1, we reanalyzed the occurrence of collapses during stepping and their relation to over-extensions.

1) Slipping and instability: Analyses of the landing angles during stepping revealed no significant differences between the manipulated and control chickens. Moreover, the angles preceding a collapse were not smaller but rather slightly higher than those that did not precede a collapse. Collectively these findings suggest that the instability does not involve over ground slipping. These analyses are shown in the new figure 7—figure supplement 2 and described on p. 34 lines 396-407, in the methods section lines 671-673.

2) Collapse and over-extension: Our analyses revealed that 65% of the collapses were not preceded by or followed by overextensions. Only 22.5% of the collapses, were followed by an overextension, and there was a high variability between chickens. In 12.5% of the cases the collapses were preceded by an overextension. These new analyses are summarized on p. 30-33 lines 358-369.

Together, the new analyses strengthen our original inferences outlined in the discussion, that the motor phenotype can be attributed to perturbed spinal information supply to the cerebellum. Furthermore, interference with the descending regulation of spinal motor output, or a lack of proper direct input to intraspinal premotor neurons. The few overextensions observed after collapses, may reflect a post-collapse compensation of the extensor drive (p. 30 lines 368-369).

2) the identity of which sub-population of neurons is considered to be responsible for the locomotor behavior (di2 neurons vs vSCT neurons),

Similar to other spinal cardinal populations of INs, dI2 neurons consistof several subpopulations. Among them, the VSCT dI2 and the propriospinal dI2. The division to these subclasses is based on the size of the neuron somata (larger versus small cell diameter), and the intra- and supra-spinal projection pattern. The lack of genetic means that enable targeting these dI2 sub-populations, hinders the analyses of their relative contribution to stable locomotion. An alternative experimental paradigm is to target specifically the VSCT dI2 by target-derived activation of neural-modulators. Namely, to activate neural modulator in dI2 neurons retrogradely from the cerebellum or from the contralateral lumbar spinal cord (injection of PRV-Cre into the cerebellum). In the original manuscript we demonstrated that dI2 axons send numerous collaterals along the entire extent of the spinal cord. The origin of these collaterals, either from the VSCT or the propriospinal dI2 neurons, could not be accurately resolved in the original manuscript. We suspected that the VSCT dI2 neurons also innervate the lumbar spinal cord, which may undermine the use of target-derived neural-modulators activation.

In order to resolve the connectivity of VSCT dI2 neurons, we employed lumbar-specific sparse labelling of dI2, and followed in a single-cell resolution, their entire lumbar to cerebellum trajectory patterns, utilizing light sheet microscopy. The new data are presented in the new figure 3, four supplementary videos, supplementary figure 3—figure supplement 1A (p. 15-16, lines 199-222). Our new data provides evidence that VSCT dI2 neurons send collaterals that innervate the contralateral targets along the entire path, from the lumbar spinal cord to the brainstem and the cerebellar nuclei.

This new analysis questions that validity of target-derived silencing of dI2 subpopulations, either via retrograde activation from the cerebellum or the contralateral lumbar spinal cord, since this approach will target both the propriospinal lumbar dI2 and the VSCT dI2 neurons. Future experiments, aimed to uncover the transcriptome of dI2, may provide genetic means to target dI2 subpopulations and subsequently to manipulate their neuronal activity. We discuss the relative potential contribution of these two subpopulations to stepping on pages 42-43 (lines 473-489).

In this regard, It should be noted that the physiological role of other spinal interneurons that control motor activity, like V1, V2, V3, V0, dI3 and dI1 (Yuengert et al., 2015, Gosgnach et al., 2006, Zhang et al., 2008, Talpalar et al., 2013, Bui et al., 2013, Crone et al., 2008), was previously studied via genetic targeting using regulatory elements of transcription factors expressed early in their development, similar to our targeting and silencing approach. Hence, our wiring and behavioral studies fall within the scope of the experimental strategies in the field.

3) the anatomical data of the neurons of interest, possibly including their neurotransmitter phenotype,

The new light sheet microscopy analyses (figures 3, figure 3—figure supplement 1A and the four supplementary videos) provide insights into this question. To complement these, we also preformed new in situ mRNA localization using vesicular inhibitory amino acid transporter (VIAAT) RNA probes to detect both GABAergic and glycinergic inhibitory interneurons. We also preformed new in situ hybridization experiments with a vGglut2 probe for detection of excitatory interneurons. The proportion of the excitatory/inhibitory dI2 neurons is 73%/27%. The new data are presented in figure 1G-I and in p.11 lines 150-153.

4) The quantification and characterization of the synaptic contacts and their potential functional role.

In the original manuscript we have used two criteria for detection of synaptic contacts to and from dI2: (1) The likelihood of connectivity was examined by spatial overlap of axonal terminals from the presumed presynaptic neurons and the somata of the post synaptic neurons. (2) Detection of synaptic boutons, labelled by a synaptic reporter, on the somatodendritic membrane of the post synaptic neurons. We also included an image showing an overlap between a synaptic reporter and a synaptic protein (former figure 3C, current figure 4C’).

In the revised manuscript we thoroughly tested the validity of the genetically-delivered synaptic reporter by quantifying the co-labelled staining of synaptic reporter with synaptotagmin. We used confocal imaging and 3D reconstruction using IMARIS software to score signal overlap. We tested the co-labelling of dI2::SV2-GFP and synaptotagmin in dI2 neurons synapses on contralateral pre-motoneurons. From 144 genetically labeled boutons, 121 (84%) were synaptotagmin^+^. The synaptotagmin^-^ boutons were significantly smaller. We set a volume threshold, so that small volume SV2-GFP boutons were not considered as synapses throughout the study (Figure 4—figure supplement 1 and p. 20 lines 244-249).

Thus, we are confident that our synaptic reporter staining coupled with the labelling of the somata of presumed target neurons, reliably represent anatomical synapses. We also would like to note that similar anatomical-synaptic analysis, are the convention in the field of spinal circuitry (Baek et al., 2019, Dougherty et al., 2013, Goetz et al., 2015, Ruder et al., 2016).

Reviewer #1:This is a well-put-together manuscript describing carefully performed circuitry dissection and functional analysis of dl2 neurons in the chick. A genetic toolbox is used taking advantage of the electroporation technique applied to the embryos. The findings include a fairly convincing connectome for dl2 neurons and a functional phenotype that is, unfortunately, rather unsatisfying. The investigators conclude that dl2 interneurons regulate "stability" of bipedal stepping in the chick, which is fine, but the analysis misses an opportunity to more fully explore what the instability involves and thus to perhaps shed more light on the likely roles of this neuron population. The concerns/issues 3 and 4 below focus on this issue and the need for additional careful analysis of the behavior that will allow the phenotype to be more precisely described or ascribed to some aspect of stepping that might guide future studies in other models. For example, can the link between partial collapse and over-extensions be made more solid and thus argue that reduced extensor gain might be what results in the instability? What other analysis could be performed using the existing data/video to better describe the behavioral phenotype?Concerns/issues:1. The connectome part of the work appears solid and supports the concept that a sub population of the population are likely VSCT neurons, that the non VSCT neurons receive the bulk of the afferent input and that these neurons project to contralateral dl2 neurons (some which may be VSCT) and other premotor neurons. Anatomically, the only concern is that no distinctions were made between the lumbar and brachial populations, and if differences in these populations exist, it would be important and interesting to describe them.

The present study was focused on the characterization of lumbar VSCT dI2 and their presumed contribution to hindlimb locomotion (see the modified title of the manuscript). We agree with the reviewer that the studies of the connectome and function of brachial populations of dI2 are important and interesting. We plan to study the brachial dI2 and the presumed role of lumbar to brachial dI2 connectivity in coordinating legs/wings movements in the near future.

2. Figure 2 Characterization of dl2/VSCT neurons as being primarily large dl2 neurons is quite convincing, and the observation that the dl2 neurons account for 10% of the VSCT axons is also of interest and quite compelling. A question arises, however, about the source, rostrocaudally, of the VSCT neurons and tract. Is the 10% for the total or for a specific level or levels? Can more be said/quantified about differences in these populations at different spinal levels?

See our response to the general comment #2 above, in pages 2-3 in this document. As stated in the manuscript “large-diameter dI2 neurons are only apparent at the lumbar level” (p. 12, lines 158).

3. Whole-body collapses and subsequent over-extensions are important and speak to changes in reflex arc and motor output. The statement "usually followed by" over-extension should be followed-up. Can this be further quantified? Are the two events linked or distinct, and did over-extensions happen in the absence of collapses?4. These issues mesh with the lower knee height and angle of the TMP joint, even when collapses are excluded. It appears as though the control system to maintain muscle shortening (force output of extensors) is altered. I agree that stability is compromised, but could we go further to state that the compromise is due to extensor gain control?

See our response to the general comment #1 above. We omitted the term “usually followed” and modified the text accordingly (p. 30-33, lines 358-369).

Reviewer #2:This work addresses the possibility that developmentally-characterized di2 neurons contribute to the ventral spinocerebellar tract and regulate stepping in the chick. The work is sound considering that most information we have on spinal subtypes are for ventrally-born and local circuit interneurons (i.e. motor related), but less is known about the dorsally-born types and about long-range projecting neurons that link the spinal cord with higher integrative centers. Here, using a combination of cell-type specific manipulations, circuit tracing tools and kinematic analysis of gaits in the chick authors propose that spinal di2 interneurons contain multiple subgroups including a population that sends projection to the cerebellum. Silencing di2 neurons overal leads to impaired stepping.Overall, the strategy is sound and there is potential novelty that make this paper a suitable candidate for eLife provided the weaknesses in the scientific demonstration listed below can be first addressed, experimentally and/or by additional analysis. Equally importantly also, the work suffers from a SEVERE lack of clarity (writing, figures, results). Both must be addressed for this paper to be considered further.I start with the scientific weaknesses:1. Synaptic connections rely mostly on the anatomical overlap between di2 cells and the synaptic field of their putative pre-synaptic partners. While this is indeed suggestive, it is not enough to ascertain actual synaptic connections, and even less so in a comparative manner between the different groups. Furthermore, some tracers (e.g PRVmCherry) do not seem to be under a synapse-specific promoter, so labelled elements might just as well be passing fibers. Clearer evidence of actual connections should be provided, functionally if possible or at the very least by showing clearer putative boutons onto neuronal somata/dendrites, quantifying them and quantifying differences between input cell types. Current figures (2F / 3B', C', D' / 4C, D', E', F') are not sufficiently convincing since we see only one cell and can barely detect boutons visually on some of them (not to mention that pseudo-colors keep changing, see other comment below). In addition, please consider using the term "putative" or "presumed" synapses, contacts and connections throughout the study.

See our response to the general comment #4 above (the quantification and characterization of the synaptic contacts and their potential functional role).

We agree with the reviewer that demonstrating functional connections is the gold standard for synaptic connectivity. However, we would like to note that the chick is not a common model organism for studying neuronal circuitry and post embryonic motor behavior in genetically manipulated embryos. Hence, many of the circuit-deciphering tools are not yet applicable to avian. The tools that we employed for decoding circuitry in the chick spinal cord, were mostly developed and implemented by us (Hadas et al., 2014). The main caveat is the non-germline targeting of specific neurons via electroporation. The efficiency of transgenesis via electroporation varies from 10-60%. Hence, the number of labelled putative synapses between two genetically-labeled neurons is markedly lower from the expected number in germline targeted mice. In addition, performance of in vitro recording from spinal cord neurons in chick, for demonstrating functional connectivity, is limited to early stages of embryonic development and are not applicable post E12 (O'Donovan et al., 1994). At this stage the network has not yet matured. Our efforts to modify the experimental conditions has not yet materialized.

As to the use of PRV-cherry to label contacts between pre-MNs and dI2, we agree with reviewer that this type of contacts is suggestive. We noted this in line 268-270. However, the use of 3D confocal imaging enables us to distinguish between a terminal and a passing-by axon. Importantly, in the following experiment we also use synaptic reporters expressed in two type of pre-MNs (dI1 and V1) to provide supportive evidence to the connection between pre-MNs and dI2.

2. The loss of function and gait analysis is stronger and convincingly presented. However, unless I missed it, the strategy silences all di2 neurons but cannot discriminate the contributions of the pre-cerebellar ones. This poses problems for the interpretation of the data. Since this paper is about either subpopulations of di2, or the vSCT (see other comment about general scope of the work), it would be more robust if more specific silencing was included. It is currently assumed that one likely mechanism for the disturbed gait owes to the function of di2 as precerebellar neurons (line 385, 389) but the phenotype could also, or even entirely, be due to their proprio-spinal connectivity. This is a major caveat.

See our response to the general comment #2 above (The identity of which sub-population of neurons is considered to be responsible for the locomotor behavior).

On top of this, writing and data presentation MUST be substantially improved on multiple aspects:3. Please have the manuscript deeply proofread. In addition to numerous English mistakes (missing "the", "or", plural and singulars, lots of unnecessary comas, etc…) examples of confused writing include (non-exhaustive list):(a) Line 128: what does this phrase mean ("TF expression is redundant"…)

We have corrected this to: “…their expression is not required after the establishment of the circuitry”. The manuscript was deeply edited.

(b) Line 159: I don't understand here, the Di2 ascend to the cerebellum, cross the midline to the targeted di2?? To which Di2 do authors refer to here, it sounds like they are in the cerebellum, or that the ascending Di2 redescend to the spinal cord…

This was a typo. The corrected sentence is “…and cross back to the other side of the cerebellum ipsilaterally to the targeted granular neurons”.

(c) The term targeted is in fact used alternatively and confusingly to refer to either "manipulated" cells, "synaptically-targeted" cells, there is also "targeted overground locomotion",….

The word targeted in the sentence: "targeted overground locomotion" is indeed redundant and was omitted. We reduced the use of “target” to indicate the putative post-synaptic neurons of dI2, throughout the manuscript. However, we didn’t eliminate it completely, since the use of “target” in conjunction to both genetic-targeting and synaptic-target is acceptable and very common in the field of spinal circuitry.

(d) Stage HH18 is sometimes referred to as E3. Please be consistent throughout.

We use now the term HH18 throughout the manuscript.

(e) When describing inputs onto di2, add "neurons" (i.e. "onto di2 neurons").

The term ״neurons״ is now added.

4. I would appreciate more background on di2 neurons in the introduction and why these have been investigated. Currently, most of this is given in the first paragraph of the results (lines 91-100 and also line 103). Also, it is stated first that "the role of di2 neurons is elusive due to the lack of genetic targeting means" (line 59). This contradicts the later statement that "the progenitor pdi2 expresses [various transcription factors]", and that the "post mitotic di2 are defined by…" (line 103). Please clarify what is known and not known about di2 already in the introduction.

A new background paragraph was added to the introduction on p. 3, lines 59-70.

5. Related to the above, it is not sufficiently clear what is investigated here. The genetic identity of ventral spinocerebellar neurons? Or the diversity of di2 neurons? In the way the introduction is written, it gives the impression that it is the former, but then functional investigations are not specific enough (since they are targeted to the overall di2 population, see dedicated comment later). Authors should revise to make clearer what is the scope of the work.

We thank the reviewer for highlighting this issue. The focus of the study is indeed dI2. We modified the introduction section accordingly. As to dI2 subpopulations, see our response to comment #2 in the summary section of the *eLife* decision letter.

6. Histology Figures should be made more convincing, self-explanatory, and to a higher standard.(a) Anatomical landmarks MUST be paced on ALL figures, e.g: the midline and minimal nuclei of the cerebellum, the deep cerebellar nuclei should be indicated in Figure S4,… Also, please give the orientation axis on ALL figures (especially the ones illustrating large territories, like 2B, 4A).

We replaced the image of the central cerebellar nuclei in figure 2—figure supplement 1A (the previous S3A). A low magnification image is now included and the midline and the margins of the central cerebellar nuclei are indicated. We are not aware to the term “minimal nuclei of the cerebellum”.

We inserted coordinates to indicate the rostral-caudal ipsi-contra and medial-lateral axes. However, we did not indicate the ventral-dorsal axis in the cross-section images, since it is acceptable that ventral is down in these images.

(b) Add the CTB or HSV tracer on Figure 2A and check coherence: I believe for instance that HSP is wrongly stated instead of HSV in Figure 2D and PRV is wrongly stated instead of CTB in Figure 2F (and there might be other confusions throughout).

In Figure 2D HSP is indeed HSV. In Figure 2F we have used PRV-cherry. We calibrated the post-injection time that is required for the infection of lumbar level pre-cerebellar neurons. We also co-injected PRV-cherry+CTB to verify that just the pre-cerebellar (not the pre-pre-cerebellar) neurons are labelled.

We amended the text accordingly on p. 15 lines 187-189.

(c) It is extremely confusing that histology pseudo-colors are sometimes changed from one related figure to the other, for unclear reasons (e.g. 2B, 2B', 2C, also 2C and S4A…). Consistency will help the reader go through all panels and figures comparatively.

We have now adjusted the colors: dI2 neurons are cyan, pre-dI2 are magenta and post-dI2 neurons are yellow.

(d) Figures must be addressed in proper order. This also applies to supplemental figures. Otherwise, it gives the impression we have missed something.

Done.

(e) What is the rationale for plotting the overlap in area versus volume (Figure 2H, I)? If overlap with area shows a higher percentage than with volume, does it mean that the overlap is only limited to a given A/P plane? I'm really confused about this representation and its meaning.

We agree that the added value in showing the overlapping histograms is marginal. Instead, we inserted a laminar distribution chart of each type of neuron and synapses, as suggested by the reviewer, in figures 1, 2, 4 and 5 and the corresponding supplemental figures. The overlap between the cell-to-cell and synapses-to-cells is also apparent in this presentation.

7. Authors should avoid relying on subjective formulations like "that reside at the lateral dorsal aspect of lamina VII". Instead, they MUST demonstrate the positioning of Di2 neurons into the different spinal laminae with some form of quantitative measurements. This is currently just an "impression" that large, precerebellar Di2 are more ventral, in lamina VII and possibly VIII but without the representation of lamina borders on figures, this information cannot be appreciated by the reader. It is all essential that these borders are depicted in Figures and neurons be quantitatively allocated to each laminae. In addition/alternatively, authors should report the average D/V position of the different subtypes and test for significant differences to make the case of different spatially-confined populations stronger.

The laminar distribution of all the neurons and synapses is now included. See figures 1,2,4 and 5 and the corresponding supp figures.

8. FoxD3 expression on Supplemental Figure 2B is not convincing. It is also not reported in the statistics of Figure 1E. Do we have to assume that all di2 investigated here are FoxD3-positive? If so, one would need a better illustration and quantifications should be given. Otherwise, I would suggest to simply rely on literature and removing that Figure S1B which is not helping. On other panels of that supplemental Figure 2, please add arrow/arrowheads on all neurons that are or not co-labelled so we can appreciate co-labelling.

As suggested by the reviewer we now show only the expression of FoxD3 in the post-mitotic pre-migratory dI2, and we provide the proper citation for Foxd3 expression. We preformed new in situ experiment and modified Figure 1—figure supplement 2B. We added arrow/arrowheads on all neurons in figure 1—figure supplement 2.

Figure 1—figure supplement 1B is informative for the general audience. Thus, we think that it is important to include it.

9. The demonstration that di2 are excitatory is essential. It is the title of a paragraph (line 102), thus I think that the corresponding data with the neurotransmitters (Vglut2, GAD) would deserve to be in the main Figures. Also, the chosen illustration only shows ONE double-labelled cell with Vglut2. Authors should be able to show a field of view that more convincingly conveys the message with more cells.

We preformed new in situ mRNA localization experiment, this time using the VIAAT probe for detecting the inhibitory dI2 neurons. The proportion of the excitatory/inhibitory dI2 is 73%/27% (see figure 1G-I).

Reviewer #3:This study by Haimson et al., aims at examining the diversity of dI2 interneurons and their role in coordinating activity across difference region of the spinal cord and in reporting back activity to the brain. The results show that dI2 interneurons comprise different sub-classes based on their axonal projections, soma diameter and transmitter identity. They also show that some dI2 interneurons project rostrally from the lumbar spinal cord and make putative synaptic contacts with other dI2 interneurons in the brachial spinal cord on their way to the cerebellum. Finally, it is shown that some dI2 interneurons receive putative inputs from DRG neurons and may serve to transmit movement-related feedback. An indiscriminate silencing of dI2 interneurons results in instability of locomotion. Overall, this study reports some interesting observations by showing the heterogeneity of dI2 interneurons and their potential function. I have the following concerns:1) 12% express Pax2 and are considered inhibitory. However, Gad is expressed in only 25% of dI2 interneurons while vGlut is expressed in 88%. These proportions suggest that there are dI2 neurons that co-express vGlut and Gad. Is this the case? Are there additional inhibitory dI2 neurons in addition to those expressing Pax2 which could explain the fact that Gad labels 25% of dI2 neurons. These points need some clarifications and discussion.

Pax2 labels GABAergic inhibitory neurons. In order the label all the inhibitory interneurons we have now employed VIAAT as a probe. See also our response to the general comment #3 above.

2) Of all dI2 interneurons, 91% are small diameter and 9% are large diameter neurons – large diameter neurons are mostly apparent in the lumbar spinal cord. The small and large diameter dI2 neurons cannot be differentiated by their expression of TFs, but can be distinguished by their transmitter identity? Is the proportion of small and large diameter neurons the same along the spinal cord?

We have now analyzed the number of inhibitory/excitatory neuron is the small and large diameter dI2 neurons. We found that the excitatory/inhibitory ratio in both subpopulation is similar. These new data are presented in Figure 1G-I and in line 150-153.

3) Do all dI2 neurons receive putative synaptic contacts from DRG neurons? Unless I have missed it, it would be helpful to provide quantification of the number of small vs large diameter dI2 neurons with regard to the different putative synaptic contacts they receive from DRG neurons, dI2 and V1 interneurons.

We have now analyzed the synaptic inputs of DRG neurons to dI2 neurons. This quantification is now presented in figure 4—figure supplement 3E,F (p. 21, lines 261-267).

4) Lines 218-220: It is stated that DRG putative contacts are mainly targeting dorsal dI2 neurons while ventral ones receive virtually no contacts. Since large diameter VSCT dI2 neurons are located ventrally, they do not seem to receive direct sensory information. However, the authors conclude that VSCT dI2 neurons receive sensory input (lines 227-228) and also in the Discussion. There seem to be a mismatch between the results and the conclusion drawn by the authors (lines 374-377). Unless I am missing something here, this is not consisting with the conclusions of this study. Please clarify.

We have now analyzed the number of contacts between DRG axons and dorsal/ventral dI2 and large/small dI2 neurons. The number of contacts with the dorsal dI2 neurons is significantly higher than with the ventral dI2. However, the number of contacts with the small and large dI2 neurons is similar. The decrease of contacts in the ventral dI2 is mainly due to fewer contacts with the small ventral dI2. These new data are presented in Figure 4—figure supplement 3E,F.

5) The silencing experiments are interesting, however it is unclear which sub-class of dI2 neurons and at what level (lumbar vs brachial spinal cord or cerebellum) the observed behavioral perturbations take place. It is possible to selectively silence excitatory vs inhibitory or only VSCT neurons to provide some link between dI2 sub-classes and behavioral perturbations.

See our response to the general comment #2 above.

References

Baek, M., Menon, V., Jessell, T. M., Hantman, A. W. And Dasen, J. S. 2019. Molecular Logic of Spinocerebellar Tract Neuron Diversity and Connectivity. *Cell Rep,* 27**,** 2620-2635 e4.

Bikoff, J. B., Gabitto, M. I., Rivard, A. F., Drobac, E., Machado, T. A., Miri, A., Brenner-Morton, S., Famojure, E., Diaz, C., Alvarez, F. J., Mentis, G. Z. and Jessell, T. M. 2016. Spinal Inhibitory Interneuron Diversity Delineates Variant Motor Microcircuits. *Cell,* 165**,** 207-219.

Bui, T. V., Akay, T., Loubani, O., Hnasko, T. S., Jessell, T. M. And Brownstone, R. M. 2013. Circuits for grasping: spinal dI3 interneurons mediate cutaneous control of motor behavior. *NEURON,* 78**,** 191-204.

Clark, A. J. And Higham, T. E. 2011. Slipping, sliding and stability: locomotor strategies for overcoming low-friction surfaces. *J Exp Biol,* 214**,** 1369-78.

Crone, S. A., Quinlan, K. A., Zagoraiou, L., Droho, S., Restrepo, C. E., Lundfald, L., Endo, T., Setlak, J., Jessell, T. M., Kiehn, O. And Sharma, K. 2008. Genetic ablation of V2a ipsilateral interneurons disrupts left-right locomotor coordination in mammalian spinal cord. *Neuron,* 60**,** 70-83.

Daley, M. A. And Biewener, A. A. 2006. Running over rough terrain reveals limb control for intrinsic stability. *Proc Natl Acad Sci U S A,* 103**,** 15681-6.

Dougherty, K. J., Zagoraiou, L., Satoh, D., Rozani, I., Doobar, S., Arber, S., Jessell, T. M. And Kiehn, O. 2013. Locomotor rhythm generation linked to the output of spinal shox2 excitatory interneurons. *Neuron,* 80**,** 920-33.

Goetz, C., Pivetta, C. And Arber, S. 2015. Distinct limb and trunk premotor circuits establish laterality in the spinal cord. *Neuron,* 85**,** 131-144.

Gosgnach, S., Lanuza, G. M., Butt, S. J., Saueressig, H., Zhang, Y., Velasquez, T., Riethmacher, D., Callaway, E. M., Kiehn, O. And Goulding, M. 2006. V1 spinal neurons regulate the speed of vertebrate locomotor outputs. *Nature,* 440**,** 215-9.

Hadas, Y., Etlin, A., Falk, H., Avraham, O., Kobiler, O., Panet, A., Lev-Tov, A. And Klar, A. 2014. A 'tool box' for deciphering neuronal circuits in the developing chick spinal cord. *Nucleic Acids Res,* 42**,** e148.

Hamburger, V. And Oppenheim, R. 1967. Prehatching motility and hatching behavior in the chick. *J Exp Zool,* 166**,** 171-203.

Hammar, I., Bannatyne, B. A., Maxwell, D. J., Edgley, S. A. And Jankowska, E. 2004. The actions of monoamines and distribution of noradrenergic and serotoninergic contacts on different subpopulations of commissural interneurons in the cat spinal cord. *Eur J Neurosci,* 19**,** 1305-16.

Hammar, I. And Maxwell, D. J. 2002. Serotoninergic and noradrenergic axons make contacts with neurons of the ventral spinocerebellar tract in the cat. *J Comp Neurol,* 443**,** 310-9.

O'donovan, M., Ho, S. And Yee, W. 1994. Calcium imaging of rhythmic network activity in the developing spinal cord of the chick embryo. The Journal of neuroscience : the official journal of the Society for Neuroscience, 14**,** 6354-69.

Provine, R. R., Sharma, S. C., Sandel, T. T. And Hamburger, V. 1970. Electrical activity in the spinal cord of the chick embryo, in situ. *Proc Natl Acad Sci U S A,* 65**,** 508-15.

Ruder, L., Takeoka, A. And Arber, S. 2016. Long-Distance Descending Spinal Neurons Ensure Quadrupedal Locomotor Stability. *Neuron,* 92**,** 1063-1078.

Talpalar, A. E., Bouvier, J., Borgius, L., Fortin, G., Pierani, A. And Kiehn, O. 2013. Dual-mode operation of neuronal networks involved in left-right alternation. *Nature,* 500**,** 85-8.

Yuengert, R., Hori, K., Kibodeaux, E. E., Mcclellan, J. X., Morales, J. E., Huang, T. P., Neul, J. L. And Lai, H. C. 2015. Origin of a Non-Clarke's Column Division of the Dorsal Spinocerebellar Tract and the Role of Caudal Proprioceptive Neurons in Motor Function. *Cell Rep,* 13**,** 1258-1271.

Zhang, Y., Narayan, S., Geiman, E., Lanuza, G. M., Velasquez, T., Shanks, B., Akay, T., Dyck, J., Pearson, K., Gosgnach, S., Fan, C.-M. And Goulding, M. 2008. V3 spinal neurons establish a robust and balanced locomotor rhythm during walking. *NEURON,* 60**,** 84-96.